# *GPTCast*: a weather language model for precipitation nowcasting

Gabriele Franch[1], Elena Tomasi[1], Rishabh Wanjari[1], Virginia Poli[2], Chiara Cardinali[2], Pier Paolo Alberoni[2], and Marco Cristoforetti[1]

[1]Fondazione Bruno Kessler, Trento, Italy
[2]Arpae Emilia-Romagna, Bologna, Italy

**Correspondence:** Gabriele Franch (franch@fbk.eu)

**Abstract.** This work introduces GPTCast, a generative deep-learning method for ensemble nowcasting of radar-based precipitation, inspired by advancements in large language models (LLMs). We employ a GPT (Generative Pre-trained Transformer) model as a forecaster to learn spatiotemporal precipitation dynamics using tokenized radar images. The tokenizer is based on a Quantized Variational Autoencoder featuring a novel reconstruction loss tailored for the skewed distribution of precipitation that promotes faithful reconstruction of high rainfall rates. This approach produces realistic ensemble forecasts and provides probabilistic outputs with accurate uncertainty estimation. The core architecture operates deterministically during the forward pass; ensemble variability arises from sampling the categorical probability distribution predicted by the forecaster during inference, rather than requiring external random inputs like noise injection common in other generative models. All forecast variability is thus learned solely from the data distribution. We train and test GPTCast using a 6-year radar dataset over the Emilia-Romagna region in Northern Italy, showing superior results compared to state-of-the-art ensemble extrapolation methods.

## 1 Introduction and prior work

Nowcasting —short-term forecasting up to 6 hours— of precipitation is a crucial tool for mitigating water-related hazards (Werner and Cranston, 2009). Sudden precipitation can result in landslides and floods, frequently compounded by strong winds, lightning, and hailstorms, which can seriously jeopardize human safety and damage infrastructure. The foundation of very short-term (up to two hours) precipitation nowcasting systems is the application of extrapolation techniques to weather radar reflectivity sequences (Bojinski et al., 2023) that ingest current and $n$ previous observations $T_{-n}, \ldots, T_{-1}, T_0$ with the aim to extrapolate $m$ future time steps $T_1, T_2, \ldots, T_m$. These short-term precipitation forecasts are essential for emergency response when released timely and communicated properly via early warning systems (Göber et al., 2023).

The main contenders to extrapolation techniques are numerical weather prediction (NWP) models, which can be used to forecast the probability and estimate the intensity of precipitation across large regions, but their accuracy is limited at smaller geographical and temporal scales (Surcel et al., 2015). Convective precipitation, which produces high rainfall rates and small cells, is especially difficult to forecast correctly for NWP models (Sun et al., 2014). For these reasons, operational weather agencies recognize the great value offered by short-term extrapolation forecasts and make heavy use of statistical and, more

recently, data-driven models that utilize the most recent weather radar observations for nowcasting (Woo and Wong, 2017; Turner et al., 2004).

Lagrangian extrapolation is the most well-known method for nowcasting precipitation (Bellon and Austin, 1978). It generates motion vectors to forecast the future direction of precipitation systems by applying optical-flow algorithms to a series of radar-derived rain fields. However, this approach becomes less accurate for increasing lead time, particularly in convective 30 situations where precipitation could increase or decrease quickly. Several alternative techniques have been studied to overcome these constraints, like the seamless integration between nowcasting and NWP forecasts (Sideris et al., 2020; Bowler et al., 2006) and the integration of orography data (Foresti et al., 2018; Panziera et al., 2011). Other, more sophisticated nowcasting methods improve the Lagrangian approach by generating ensemble nowcasts and preserving the precipitation field's structural characteristics. These sets of multiple forecasts aid in the assessment of forecast uncertainty by presenting multiple future 35 scenarios. The most widespread example of this approach is the Short-Term Ensemble Prediction System (STEPS) (Bowler et al., 2006; Seed et al., 2013).

The most recent advancements in nowcasting precipitation have seen the application of data-driven methods and, more prominently, of Deep Neural Networks (DNNs) and Generative AI techniques to enhance forecast accuracy and realism. Deterministic DNNs have been instrumental in predicting the dynamics of precipitation, including its development and dissipation, 40 overcoming one of the major shortcomings of extrapolation methods (Shi et al., 2015; Agrawal et al., 2019; Wang et al., 2018; Franch et al., 2020; Ayzel et al., 2020). However, deterministic models tend to produce less precise forecasts over time due to increasing uncertainty that manifests itself as a forecast field that smooths progressively with the lead time. Similarly to Lagrangian extrapolation, to overcome this limitation, ensemble deep learning methods have been introduced. Generative methods have significantly improved the generation of realistic precipitation fields beyond deterministic average predictions. 45 The forefront of this technology is embodied in models that employ techniques such as Generative Adversarial Networks (GANs) (Zhang et al., 2023; Ravuri et al., 2021), that enable more accurate and detailed precipitation forecasts by learning to mimic real weather patterns closely, and more recently by Latent Diffusion models (Leinonen et al., 2023; Gao et al., 2023), that can not only generate realistic rainfall forecasts but also produce reliable ensembles that can provide accurate uncertainty quantification of future scenarios. Many of these techniques were originally born in the field of computer vision and have 50 subsequently been adapted to the weather forecasting domain with resounding success (Goodfellow et al., 2014; Rombach et al., 2022).

In this study, we take inspiration from the successful trend of applying Large Language Models (LLMs) architectures (Vaswani et al., 2017; Wolf et al., 2020) born in the field of Natural Language Processing (NLP) to other disciplines (Dosovitskiy et al., 2020; Liu et al., 2021), including the medium range weather forecasting domain (Lang et al., 2024; Lessig et al., 2023), 55 intending to transfer this knowledge to the nowcasting domain. To do so, in our work, we follow a strategy that mimics the setup of natural language processing: a tokenization step, where an input tokenizer splits and maps the input to a finite vocabulary, and an autoregressive model trained on the tokens produced by the tokenizer. We show that such an approach produces realistic and reliable ensemble forecasts. Given the different characteristics of our input data compared to LLMs (i.e., spatiotemporal precipitation fields vs. texts or images), our adaptation introduces several novel contributions instrumental to our task.

## 2 GPTCast model architecture

There are two main components of our approach, which we call GPTCast:

- **Spatial tokenizer (VQGAN)**: An image compression and discretization model that learns to map patches of the radar image from/to a finite number of possible representations (tokens). The learned codebook of tokens can be used to express a compact representation of any precipitation field. The tokenizer thus has a dual role: learning how to compress and decompress the information in the input image and how to discretize the compressed information (i.e., learn an optimal codebook).

- **Spatiotemporal forecaster (GPT)**: A model trained on token sequences to causally learn the evolutionary dynamics of precipitation over space and time. Given a tokenized spatiotemporal context (a compressed precipitation sequence), the model outputs probabilities over the fixed codebook for the next expected token for the context. The output probabilities can be leveraged for ensemble generation.

This dual-stage architecture is an adaptation of the work of Esser et al., which we repurposed from the task of image generation to the task of precipitation nowcasting by introducing two key modifications:

- In the spatial tokenizer (VQGAN) model, we replace the standard reconstruction loss (MAE) with a specific loss that helps improve the reconstruction of precipitation patterns (Magnitude Weighted Absolute Error, MWAE). Moreover, the new loss also shows a promotion of the token utilization rate, where we achieve 100% codebook utilization.

- The token sequences used to train the GPT model represent a fixed three-dimensional context of time x height x width of precipitation patterns. This allows the model to learn spatiotemporal dynamics of the evolution of radar sequences.

The two components of the model are trained independently in cascade, starting with the tokenizer. This deliberate dual-stage architecture is crucial for achieving stable training and unlocking desirable properties for operational nowcasting run by meteorological services. Indeed, training the VQGAN and the GPT simultaneously with an end-to-end approach would introduce significant instability. As a probabilistic sequence model, the GPT relies on a fixed, finite vocabulary for stable operation: attempting to learn the token representation (vocabulary) concurrently with the complex spatiotemporal dynamics would force the GPT to learn dependencies over a constantly evolving vocabulary, likely hindering convergence. Furthermore, the fundamentally different architectures (CNN-based VQGAN with its specific loss functions versus the autoregressive Transformer GPT) and the challenges of backpropagation through the VQGAN's discrete quantization step would exacerbate training instability. By first establishing a robust and fixed vocabulary through the VQGAN, we create a stable foundation for the GPT to learn the spatiotemporal dynamics of precipitation. This separation allows for specialized and stable optimization of each component, ultimately enabling both realistic ensemble generation and accurate uncertainty estimation at the spatiotemporal (token) level, which are instrumental in meeting the requirements of operational nowcasting systems run by meteorological services.

Another notable feature of GPTCast is that its core architecture operates deterministically, meaning it does not require stochastic elements like injected noise during the forward pass for either training or inference. This contrasts with models like GANs or Diffusion Models (Ravuri et al., 2021; Leinonen et al., 2023; Zhang et al., 2023), which often rely on random inputs to generate variability. In GPTCast, variability for ensemble generation stems from the learned data patterns: the tokenizer learns a discrete representation, allowing the forecaster to output a categorical probability distribution over the token vocabulary for each prediction step. Sampling from this distribution during autoregressive inference generates diverse ensemble members, ensuring all variability originates from the learned conditional probability of future states given the past, rather than external randomness. (Note: Standard stochasticity in parameter initialization and optimization, e.g., SGD, is still employed during training.)

We describe the details of the model setup and novel contributions in the following subsections.

## 2.1 Spatial tokenizer: VQGAN

The spatial tokenizer is a Variational Quantized Autoencoder featuring an adversarial loss (VQGAN) (Esser et al., 2021) and a novel reconstruction loss specifically tailored to improve the reconstruction of precipitation. We carefully tune the architecture of the VQGAN to obtain a model that provides the highest possible compression, while maintaining a good reconstruction performance and computational complexity. The architecture of the tokenizer is visually summarized in Figure 1.

The encoder ($E$) and decoder ($G$) of the autoencoder are symmetric in design and formed mainly by convolutional blocks, with $\alpha = 4$ steps of downsampling and upsampling, respectively. With this setup, each latent vector at the bottleneck summarises a patch of $2^\alpha = 2^4 = $ 16x16 pixels of the input image. Following recent studies (Yu et al., 2022), we find it useful to set a number of channels at the bottleneck (i.e., the length of the latent vector) of 8 to obtain efficient utilization of the codebook, good training stability and the effective capture of essential features in a reduced-dimensional space. This choice was informed by the cited literature and our preliminary experiments, indicating a good balance between codebook utilization, training stability, and feature capture. The latent vectors at the bottleneck are discretized using a quantization layer that maps them to a finite codebook ($Z$) by finding the closest vector in the codebook. We define a codebook size of 1024 tokens in the quantization layer. The codebook vectors are initialized randomly and then learned during training.

As an example, with an input precipitation map of 192x192 pixels with a dynamic range of 601 possible values for each pixel (from 0 to 60dBZ with a 0.1dBZ step, as described later in Table 2), the resulting feature vector at the bottleneck will have a dimensionality of 12H x 12W x 8 channels. Each 8-channel vector is then mapped to one of the possible 1024 vectors in the codebook, resulting in a compressed and discretized representation of 12H x 12W with a dynamic range of 1024 values. The resulting total compression ratio of the spatial tokenizer is $\frac{192 \cdot 192 \cdot 601}{12 \cdot 12 \cdot 1024} \approx 150$ times.

To support such a high compression ratio while maintaining good reconstruction ability, especially for the extreme values, we developed a novel reconstruction loss that we use in place of commonly used reconstruction losses ($l_1$ or $l_2$, a.k.a. Mean Absolute Error or Mean Squared Error), defined with the following equation (1):

$$\text{MWAE}(\mathbf{x}, \mathbf{y}) = \sum_{i=1}^{n} |\sigma(x_i) - \sigma(y_i)| \cdot \sigma(x_i) \tag{1}$$

where $\sigma$ is the sigmoid function $\sigma(z) = \frac{1}{1+e^{-z}}$ and $x$ and $y$ are the input and output vectors of the autoencoder, respectively. We call this loss Magnitude Weighted Absolute Error (MWAE). By giving more weight to pixels with higher rain rates (magnitude), this loss simultaneously serves two purposes: the first is to nudge the tokenizer towards reserving more learning capacity for the reconstruction of extremes, and the other is to help to rebalance the notoriously skewed distribution of precipitation data, that by nature leans towards low rain rates. While the sigmoid function can saturate for very large input values, potentially diminishing the sensitivity to differences in extreme rain rates, this effect is mitigated by our data preprocessing. The input radar reflectivity values (0-60 dBZ) are linearly rescaled to the range $[-1, 1]$ before being fed into the VQGAN. Within this range, the sigmoid function operates in a quasi-linear manner, ensuring that the absolute difference term $|\sigma(x_i) - \sigma(y_i)|$ appropriately reflects differences between the scaled true and reconstructed values, even for high rain rates within the considered 0-60 dBZ range. The primary reason for using the sigmoid, rather than a purely linear weighting, is to provide robustness against potential out-of-range predictions from the decoder during training, which can occur due to the perturbations introduced by adversarial training. The sigmoid gracefully handles such out-of-range values without assigning excessively large loss values, thereby improving training stability.

Alongside MWAE and the adversarial loss, the model incorporates the Learned Perceptual Image Patch Similarity (LPIPS) loss (Zhang et al., 2018), as shown in Figure 1, which further encourages perceptually realistic reconstructions by comparing feature activations in a pre-trained network. In our preliminary experiments, while not affecting the final reconstruction performance, this loss term enabled a faster model convergence.

The interactions between loss terms during training follow the original VQGAN implementation (Esser et al., 2021). The total size of the VQGAN model is 90M trainable parameters.

## 2.2 Spatiotemporal forecaster: GPT

Similarly to Esser et al., the core predictive component of GPTCast is an autoregressive Transformer model based on the GPT-2 architecture (Radford et al., 2019). We chose this specific architecture as it represents a well-established, robust, and widely understood foundation, allowing us to focus on the novel application of the tokenization and autoregressive generation paradigm to radar nowcasting, rather than optimizing for the latest Transformer variants. GPT-2 provides a strong baseline whose components are readily adaptable for spatio-temporal forecasting tasks.

The GPTCast Transformer utilizes 24 layers and 16 attention heads, resulting in a total of 304 million trainable parameters for this forecasting component. When combined with the VQGAN tokenizer (approximately 90 million parameters, see Section 2.1), the entire GPTCast system comprises roughly 394 million parameters. While potentially smaller than the largest models currently used in natural language processing, this scale is substantial within the atmospheric sciences. For context, it exceeds the size of ECMWF's operational AI Forecasting System (AIFS, approx. 253M parameters according to its public checkpoint (Lang et al., 2024)), is comparable to recent diffusion models for dynamical downscaling (e.g., approx. 300M parameters in Tomasi et al. (2025)), and is significantly larger than prominent graph-based models like GraphCast (36.7M parameters, Lam et al. (2023)). This highlights that GPTCast, despite using an established architecture, represents a large-scale deep learning approach for precipitation nowcasting. While GPT-2 serves as an effective proof-of-concept, future work

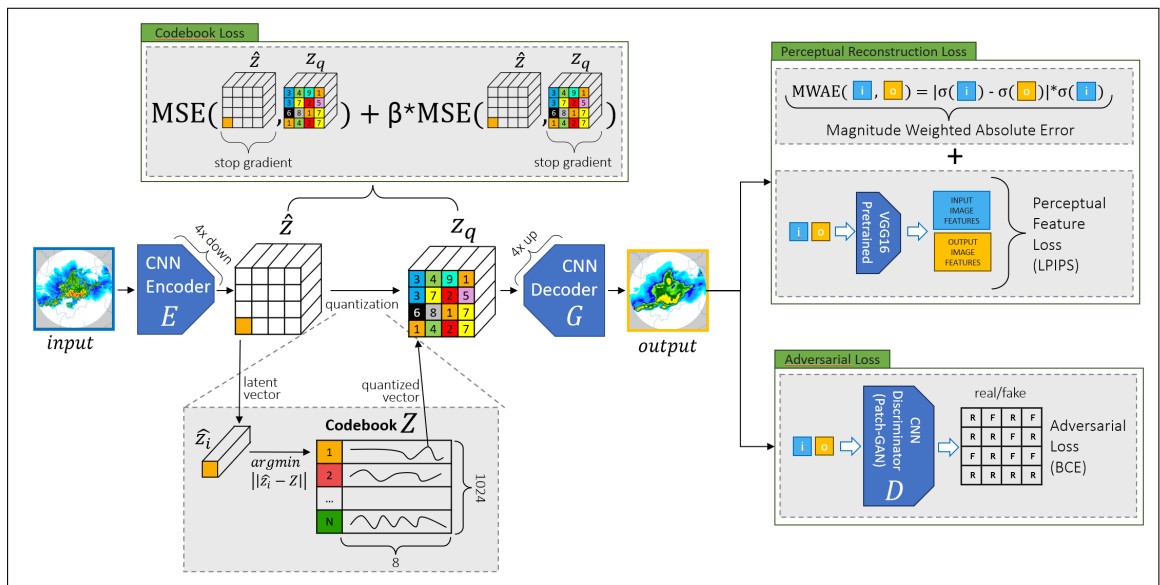

**Figure 1.** The spatial tokenizer architecture. The four loss terms (MWAE reconstruction loss, Adversarial loss, LPIPS perceptual loss, and Codebook Loss) are enclosed in boxes with green borders. The blue square [$i$] is the input image, the yellow square [$o$] is the reconstructed autoencoder output. The codebook loss is formed by two complementary parts: the gradient from the detached $z_q$ primarily updates the codebook weights, while the gradient involving the detached $z$ primarily updates the encoder layers preceding the quantization, encouraging them to produce easily quantizable representations.

could certainly explore the potential benefits of more recent or specialized Transformer architectures (e.g., those optimized for efficiency or long-context modeling) for this task.

We train two configurations, one with a spatiotemporal context size of 8 timesteps (40 minutes) x 256 x 256 pixels and a second configuration with 8 timesteps x 128 x 128 pixels. At the token level the two configurations amount to a context length of 2048 (8 x 16 x 16 tokens) and 512 (8 x 8 x 8 tokens) respectively. We refer to the two models as GPTCast-16x16 and GPTCast-8x8 respectively. In a GPT-like Transformer model, the context size (or sequence length) does not affect the number of parameters; instead, it influences the computational complexity and memory requirements of the model during training 165    and (more crucially) inference. For these reasons, careful consideration in balancing computational complexity and model performance should be made, since timely forecasts are crucial for nowcasting. A summary of the two GPT models' settings is reported in Table 1.

The training process of the forecaster is schematized in Figure 2: contiguous spatiotemporal sequences of radar data are retrieved from the training dataset and encoded into codebook indices through the frozen VQGAN encoder and passed to the 170    GPT model as training samples. The GPT forecaster is trained autoregressively to predict the probability distribution for each token $z_t$ given the sequence of preceding tokens $z_{<t}$. The tokens are ordered starting with the oldest image using a row-first

**Table 1.** GPTCast Model Configurations with large and small spatial domain

| Configuration/Model name | GPTCast-16x16 | GPTCast-8x8 |
|---|---|---|
| Vocabulary Size | 1024 | 1024 |
| Context Length | 2048 (8T x 16H x 16W tokens) | 512 (8T x 8H x 8W tokens) |
| Number of Layers | 24 | 24 |
| Number of Heads | 16 | 16 |
| Embedding Dimension | 1024 | 1024 |

format. The ordering is instrumental to the nowcasting task: in inference, we can provide the model a context that is pre-filled with the past 7 time steps to generate the tokens for the 8th time step.

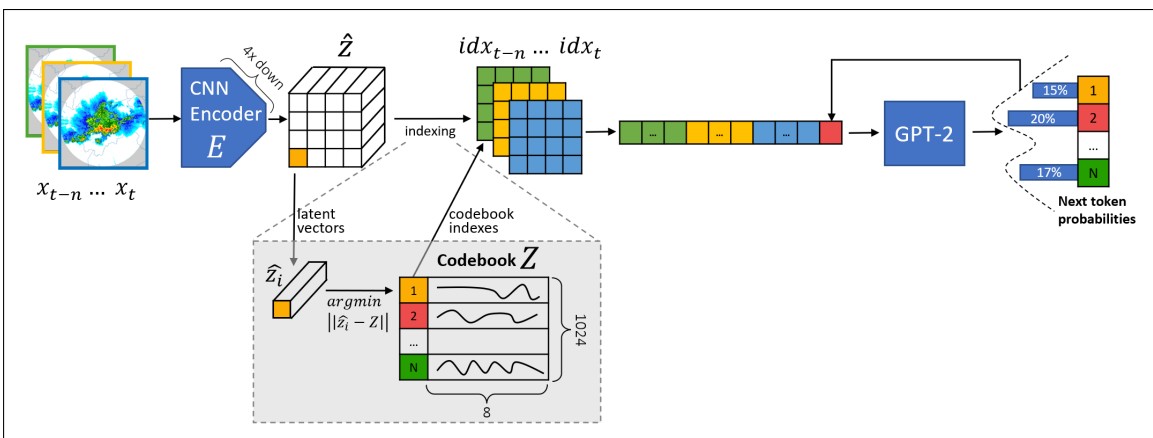

**Figure 2.** The spatiotemporal forecaster architecture. During the training of the forecaster, the tokenizer encoder ($E$) weights are frozen.

## 2.3 Inference

At inference time, the two models are combined in a sandwich-like configuration, with the encoding of the context input images through the VQGAN encoder, the autoregressive generation of the indices of multiple forecast steps via the transformer model, and the final decoding of the tokens back to pixel space using the VQGAN decoder (see Figure 3). To obtain multiple ensemble members, the autoregressive generation of the indices can be repeated multiple times while applying a multinomial draw over the output probabilities to pick different tokens.

To generate forecasts for spatial domains larger than the specific training context size, we employ a sliding window inference strategy, illustrated in Figure 4 and detailed in Algorithm 1. We process the target forecast frame sequentially, following the row-first raster scan order. To predict the token index $z_{i,j}$ for a specific spatial location $(i, j)$ in the forecast frame, we construct an input context sequence for the transformer. This sequence comprises relevant tokens from previous time steps within a

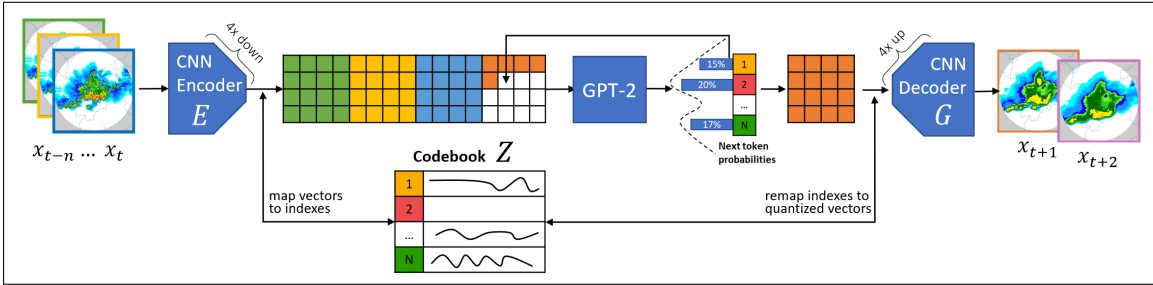

**Figure 3.** The GPTCast architecture during inference. The trained tokenizer and forecaster are combined (Tokenizer encoder ($E$) -> Forecaster -> Tokenizer decoder ($G$)) to generate forecasts. In the standard unconditional setting, the next token is chosen by applying a multinomial draw over the codebook probabilities to generate different ensemble members.

defined spatiotemporal window around $(i, j)$, along with any tokens already predicted in the current forecast frame that precede $(i, j)$ in the row-first sequential order. The transformer then predicts the probability distribution for the next token based on this context. Sampling from this distribution yields the predicted token $z_{i,j}$. This sequential, conditioned generation ensures that spatial and temporal consistency is learned and maintained across the domain via the transformer's attention mechanism, as each token prediction depends on its previously generated neighbors in space and time. The handling of domain edges occurs naturally as the available context within the sliding window adapts based on the target token's position.

## 3 Dataset

The dataset we propose for the study is the radar reflectivity composite produced by the HydroMeteorological Service of the Regional Agency for the Environment and Energy of Emilia-Romagna Region in Northern Italy (Arpae Emilia-Romagna). The agency operates two Dual-polarization C-Band radars in the area of the Po Valley, located respectively in Gattatico (44°47'27"N, 10°29'54"E) and San Pietro Capofiume (44°39'19"N, 11°37'23"E). The scanning strategy allows coverage of the entire Region every 5 minutes. The area is characterized by a complex morphology and it spans from the flat basin of the Po valley in the north to the upper Apennines in the south, and from the Ligurian coast in the west to the Adriatic Sea in the east. For the purpose of this work, scans with a radius of 125 km were chosen with a total coverage of 71172 km$^2$, summarized in Figure 5.

Arpae fully manages both the radar acquisition strategy and the data processing pipeline. They include several stages of data quality control and error correction developed to reduce the effect of topographical beam blockage, ground clutter, and anomalous propagation (Fornasiero et al., 2006). Specific corrections are applied over the vertical reflectivity profile to improve precipitation estimates at the ground level (Fornasiero et al., 2008). While these quality controls mitigate major issues, residual errors inherent to radar measurements are still be present, affecting also the corresponding Quantitative Precipitation Estimation (QPE). No rain-gauge correction is applied given the challenges of reconciling the two sources at the short integration time of 5 minutes.

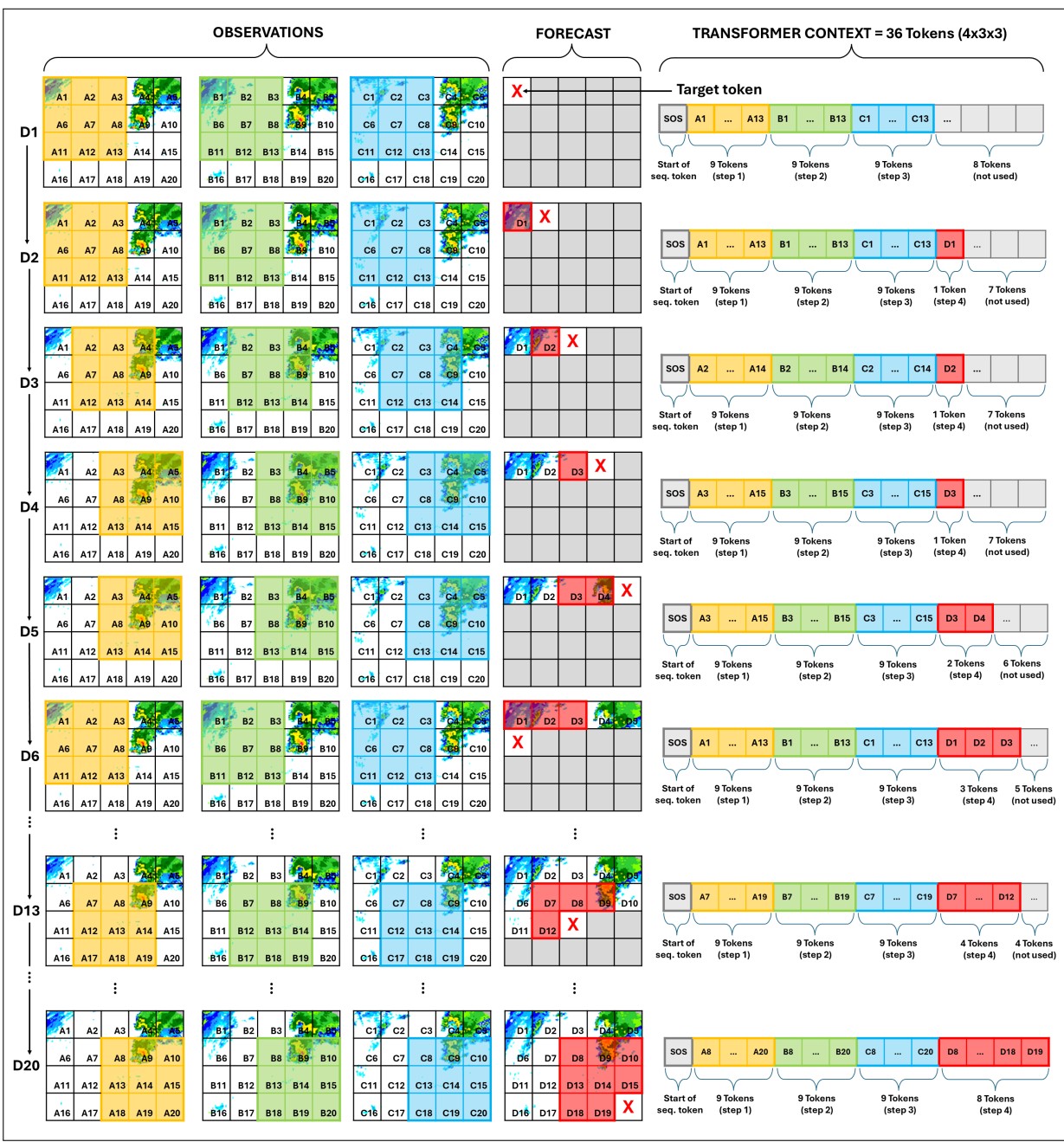

**Figure 4.** An illustration of the sliding window approach for a forecaster trained with a context length of 4 steps X 3 height X 3 width (36 tokens). Forecasts for domains of arbitrary sizes can be generated by moving the context window across the forecasting domain to predict a target token in the larger domain (starting with the token at the top left position). A fixed start-of-sequence token (index 0) is prepended to the context to provide an initial conditioning for the first token

---

**Algorithm 1** Pseudocode for Sliding Window Prediction Algorithm

---

**Require:** input_indices {Tensor of shape $[B, S, H, W]$}

**Require:** c_indices {Conditioning tokens (Start of Sequence)}

**Require:** window_size {Size of sliding context window}

**Ensure:** predicted_indices {Next frame token indices}

  $B, \_, H, W \leftarrow$ shape(input_indices)

  half_window $\leftarrow \lfloor$window_size$/2\rfloor$

  predicted_indices $\leftarrow$ Tensor$(B, H, W)$ filled with $-1$

  conditioning $\leftarrow$ reshape(c_indices) {Flatten conditioning}

  **for** $i = 0$ **to** $H - 1$ **do**

    **for** $j = 0$ **to** $W - 1$ **do**

      /* Calculate window boundaries with edge handling */

      $i_{\text{start}} \leftarrow \max(0, i - \text{half\_window})$

      $i_{\text{end}} \leftarrow \min(H, i_{\text{start}} + \text{window\_size})$

      $i_{\text{start}} \leftarrow \max(0, i_{\text{end}} - \text{window\_size})$ {Adjust if at bottom edge}

      $j_{\text{start}} \leftarrow \max(0, j - \text{half\_window})$

      $j_{\text{end}} \leftarrow \min(W, j_{\text{start}} + \text{window\_size})$

      $j_{\text{start}} \leftarrow \max(0, j_{\text{end}} - \text{window\_size})$ {Adjust if at right edge}

      /* Extract past context and already predicted tokens */

      past_tokens $\leftarrow$ flatten(input_indices$[:, :, i_{\text{start}} : i_{\text{end}}, j_{\text{start}} : j_{\text{end}}]$)

      pred_patch $\leftarrow$ predicted_indices$[:, i_{\text{start}} : i_{\text{end}}, j_{\text{start}} : j_{\text{end}}]$

      window_pos$_i \leftarrow i - i_{\text{start}}$

      window_pos$_j \leftarrow j - j_{\text{start}}$

      tokens_count $\leftarrow$ window_pos$_i \times (j_{\text{end}} - j_{\text{start}}) +$ window_pos$_j$

      pred_tokens $\leftarrow$ first tokens_count elements from flattened pred_patch

      /* Build context and predict next token */

      context $\leftarrow$ concatenate(conditioning, past_tokens, pred_tokens)

      next_token $\leftarrow$ predict_next_index(context)

      predicted_indices$[:, i, j] \leftarrow$ next_token.squeeze() {Fix shape mismatch}

    **end for**

  **end for**

  **return** predicted_indices

---

The resulting product is a 2D reflectivity composite map on a 290 x 373 km grid at a resolution of 1 km$^2$ per pixel, with a time step of 5 minutes. The data is provided in dBZ units (Reflectivity factor) with original values in the range from -20 dBZ to 60 dBZ. To further minimize the presence of spurious echoes and drizzle, the reflectivity values are clipped between the range of 0 and 60 dBZ, where 0 dBZ represents no precipitation and 60 dBZ a rain-rate of 205 mm/h (the radar saturation

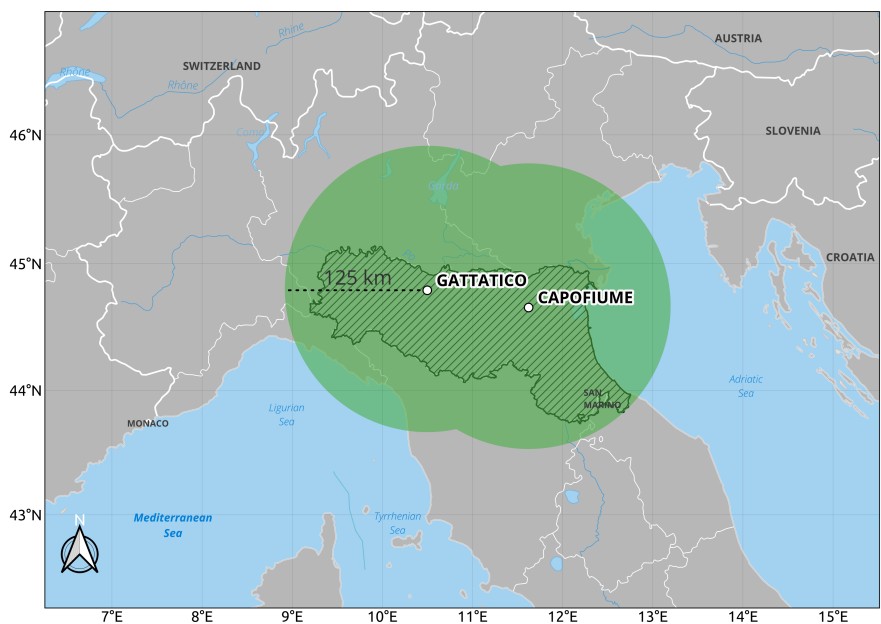

**Figure 5.** Extent of the dataset. Effective coverage is the composite of the 125 km range of the Gattatico and San Pietro Capofiume radars (green area). Hatched area is the Emilia-Romagna Region.

point). The conversion from dBZ to rain-rate is done by applying the standard Marshall-Palmer Z-R (Marshall and Palmer, 1948) transformation with parameters $a = 200$ and $b = 1.6$.

### 3.1 Data selection, preprocessing and augmentation

For the purposes of our study, we extract all contiguous precipitating sequences in the 6 years between 2015 and 2020. Non-precipitating sequences are discarded, resulting in the selection of 179,264 timesteps out of 630,720 (71,5% of the data is discarded). Specifically, we remove all timesteps where the average precipitation over all pixels in the entire domain is less than 0.01 mm/h for at least one hour. The remaining sequences are retained only if they form a contiguous sequence of at least 3 hours. This focus on precipitating events aims to concentrate the model's learning on the complex dynamics of precipitation itself. The handling of non-precipitating inputs, which are common in operational scenarios, is discussed further in Section 5 and addressed empirically in Section 4.2.5 where we test the model's behavior with entirely non-precipitating synthetic inputs.

The precipitating sequences are divided between training, validation, and test sets, and the data values are preprocessed by rounding the values to the first decimal digit, resulting in an effective dynamic range of 601 values (from 0 to 60 with a 0.1 step) per pixel.

We prepare two test sets, one for the testing of the spatial tokenizer and one for the testing of the forecaster. To test the spatial tokenizer we isolate all time steps belonging to the days in the years 2019 and 2020 where extreme events happened

by analyzing historical weather reports, resulting in a total of 21,871 radar images (time steps). We call this the *Tokenizer Test Set* (TTS). To test the forecaster we follow the same validation approach of Pulkkinen et al., and we extract out of the TTS 10 sequences of 12 hours each representative of the most relevant events. This 120-hour subset, namely the *Forecaster Test Set* (FTS), is used for the testing of the forecaster.

The remaining sequences are randomly divided between training and validation, with the following final result: 149,524 steps for training, 7,869 for validation, and 21,871 for the TTS that includes 1,450 steps (12 hours * 10 events) of the FTS. To further increase the training dataset size and promote generalization, we apply random cropping, random 90-degree rotation and flipping to the training dataset during the training phase. The primary motivation for this augmentation strategy is pragmatic: to increase the effective size and variability of the training dataset and, crucially, to mitigate overfitting. We observed, particularly for the larger GPTCast-16x16 model, that training without augmentation led to overfitting on the validation set relatively early. Introducing these random transformations allows for significantly longer training periods, improving the model's generalization by encouraging invariance to the orientation of precipitation features.

We acknowledge that this approach has trade-offs. By making the dataset invariant to orientation, we prevent the model from explicitly learning geographically fixed patterns, such as precipitation enhancement due to specific orography or effects related to dominant wind directions within the fixed geographical domain. We do not provide additional contextual information (e.g., topography, large-scale wind fields) to the model, partly to maintain a fair comparison with the baseline extrapolation methods (introduced in Section 4.2.1), which also operate solely on the precipitation fields. The chosen augmentation strategy therefore prioritizes learning the inherent dynamics, structure, and evolution of precipitation patterns themselves, aiming for a model that generalizes well to these dynamics regardless of their orientation within the frame, at the expense of capturing location-specific effects.

Table 2 summarizes the resulting dataset characteristics.

**Table 2.** Summary of dataset characteristics

| Attribute | Details |
|---|---|
| **Product Description** | Arpae radar reflectivity composite (northern Italy) |
| **Map Size** | 290 x 373 pixels |
| **Pixel Size** | 1km resolution |
| **Timestep** | 5 minutes |
| **Reflectivity Range** | -20–60 dBZ (clipped to 0-60 dBZ, 0.1 step = 601 values of dynamic range) |
| **Date range** | precipitation sequences in the years 2015 - 2020 |
| **Dataset size** | 630,720 total timesteps (179,264 precipitating timesteps selected) |
| **Train and validation** | 149,524 timesteps for training, 7,869 validation |
| **Test datasets** | *TTS*: 21,871 timesteps, *FTS*: 1450 timesteps (10 events of 12 hours) |

## 4  Results

Before presenting the quantitative and qualitative results, we clarify the roles of the different data subsets used throughout model development and evaluation.

All model development, hyperparameter tuning, and selection processes were performed using only the training and validation sets. This includes the selection of the final VQGAN tokenizer architecture (based on reconstruction fidelity and downstream performance on the validation set, comparing MAE and MWAE variants) and the selection of the best-performing GPTCast forecaster checkpoint (based on metrics evaluated exclusively on the validation set).

The two test sets (FTS and TTS) were used for the final evaluation presented in the following sections, after all model architectures and checkpoints were finalized based on validation performance. To further assess generalization to truly independent data beyond the scope of the original dataset, we also present an evaluation on a separate, out-of-distribution dataset over Germany in Section 4.2.4.

We analyze the performance of our model at two stages: first, we analyze the amount of information loss introduced by the data compression in the tokenizer, and then we analyze the performance of GPTCast as a whole for the nowcasting of precipitation up to two hours in the future. All scores and measures in the result section are computed on rain-rate values (after applying Z-R conversion).

### 4.1  Spatial tokenizer reconstruction performance

Given the high compression ratio that we introduce in the VQGAN it is crucial to understand how much and what type of information is lost during the compression and discretization step operated by the tokenizer. Depending on the nature of the information loss, certain phenomena may be completely lost, and this can compromise the ability of the transformer to learn and forecast some precipitation dynamics (e.g., extreme events). The new MWAE loss introduced in Section 2.1 is specifically built to improve the reconstruction performance of the tokenizer and reach a good level of data reconstruction while maintaining a high compression factor.

Table 3 shows the performance in reconstruction ability on the TTS between a VQGAN trained using as reconstruction loss a standard Mean Absolute Error (MAE) and using our proposed MWAE loss. We consider both global regression scores like Mean Absolute Error (MAE), Mean Squared Error (MSE), and the Structural Similarity Index Measure (SSIM, (Wang et al., 2004)) along with categorical scores computed by thresholding the precipitation at multiple rain rates (1, 10 and 50 mm/h), like the Critical Success Index (CSI) and the frequency bias (BIAS).

The autoencoder trained with MWAE shows significant improvements over all the considered metrics, but it is crucial to notice that the improvements are more pronounced for higher rain rates, whose frequency is almost precisely reconstructed by the autoencoder. This is clearly visible in the improvements in BIAS at 50mm/h, which is defined as the fraction between the number of pixels in the input image over 50 mm/h and the number of pixels that surpass the same threshold in the reconstruction, where we obtain a jump in performance from 0.22 to 0.92 (where 0 is total underestimation, 1 is the perfect score, and greater than 1 is overestimation).

**Table 3.** Reconstruction performance on the TTS of VQGAN trained with Mean Absolute Error (MAE) loss and with our proposed MWAE loss. (↓) means lower is better, (↑) means higher is better, and for frequency bias (BIAS) closer to 1 is better. The best model is in bold.

| model / performance | MAE (↓) | RMSE (↓) | SSIM (↑) | CSI (↑) / BIAS @ 1 $mm^{-h}$ | CSI / BIAS @ 10 $mm^{-h}$ | CSI / BIAS @ 50 $mm^{-h}$ |
|---|---|---|---|---|---|---|
| **VQGAN MWAE** | **0.204** | **2.02** | **0.988** | **0.81 / 1.03** | **0.56 / 0.94** | **0.44 / 0.92** |
| **VQGAN MAE** | 0.265 | 2.66 | 0.981 | 0.74 / 0.93 | 0.38 / 0.62 | 0.13 / 0.22 |

The recovery in frequency is also confirmed by analyzing the radially averaged power spectral density (i.e., the amount of energy) of the input and reconstruction: as shown in Figure 6, the average power spectra of the MWAE autoencoder closely resembles the input (albeit with an overestimation at the smallest wavelengths), while the standard autoencoder distribution is constantly shifted and underestimated at all wavelengths.

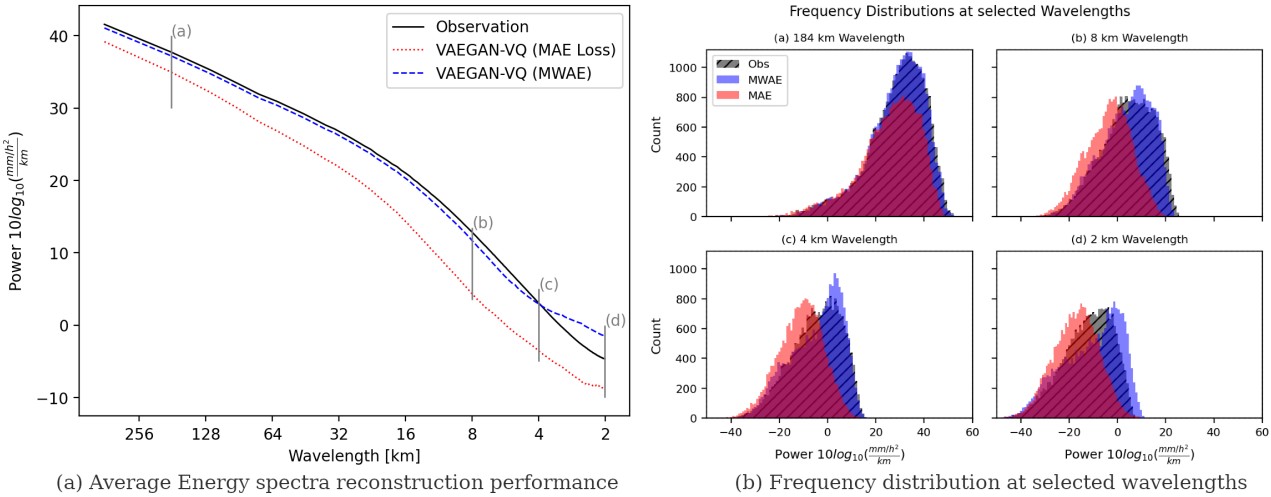

(a) Average Energy spectra reconstruction performance    (b) Frequency distribution at selected wavelengths

**Figure 6.** Comparison of radially averaged power spectral density reconstruction performance by adopting the MWAE loss function compared to MAE. The adoption of MWAE improves the ability of the autoencoder to reproduce the energy distribution of precipitation at all wavelengths.

Improvement in CSI score is also significant (at 50 mm/h, more than three times higher), albeit not as thorough as the frequency recovery. This implies that the remaining source of error is that the reconstructed precipitation fields have either a different structure or a different location when compared to the input (i.e., the amounts of the reconstructed precipitation are correct but misplaced at the spatial level).

To better characterize this remaining source of error, we compute the SAL measure (Wernli et al., 2008, 2009), which evaluates three key aspects of the precipitation field within a specified domain: structure (S), amplitude (A), and location (L). The amplitude component (A) measures the relative deviation of the domain-averaged reconstructed precipitation amount from

the input. Positive values indicate an overestimation of total precipitation, while negative values indicate an underestimation. The structure component (S) assesses the shape and size of predicted precipitation areas. Positive values occur when these areas are too large or too flat, while negative values indicate that they are too small or too peaked. The location component (L) evaluates the accuracy of the predicted location of precipitation. It combines information about the displacement of the reconstructed precipitation field's center of mass compared to the input and the error in the weighted average distance of the

precipitation objects from the center of the total field. Perfect forecasts result in zero values for all three components, indicating no deviation between input and reconstructed precipitation patterns.

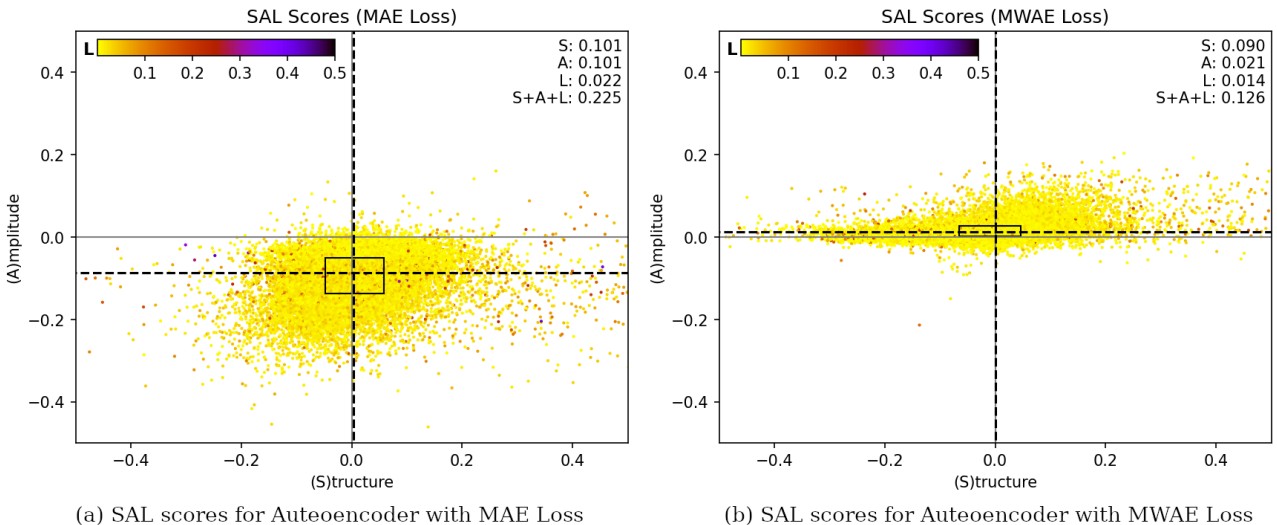

(a) SAL scores for Auteoencoder with MAE Loss      (b) SAL scores for Auteoencoder with MWAE Loss

**Figure 7.** Structure, Amplitude, and Location (SAL) plot that compares the performance of the MAE and MWAE autoencoders. Each dot on the plot represents the scores of one image in the TTS. Structure and amplitude are plotted on the horizontal and vertical axes, respectively, while the location component is represented by the color. The dashed vertical and horizontal lines indicate the median values of the Structure (S) and Amplitude (A) scores, respectively. The rectangle box represents the area between the 25th and 75th percentiles (i.e., the vertical and horizontal sides of the box contain 50% of the points). The numbers on the top right show the Mean Absolute values.

    The SAL analysis plot for both autoencoders is shown in Figure 7. The MWAE autoencoder improves over the baseline autoencoder on all scores, with a median value that is close to zero for all three components. A residual source of absolute error remains in the Structure component, while both Amplitude and Location errors are negligible.

In summary, divergences in the size and shape of the reconstructed precipitation patterns account for the majority of the error for our new autoencoder, while the locations, frequencies, and energy contents of the precipitation patches are mostly accurate. Overall, this is a good compromise for the nowcasting task since we can tolerate higher compromises for errors in structure, whereas systematic errors in amplitude, frequency, or location can seriously impair the forecaster's ability to accurately predict the evolutionary dynamics of precipitation. Some qualitative examples of the input and reconstruction from both autoencoders

are presented in Figure 8.

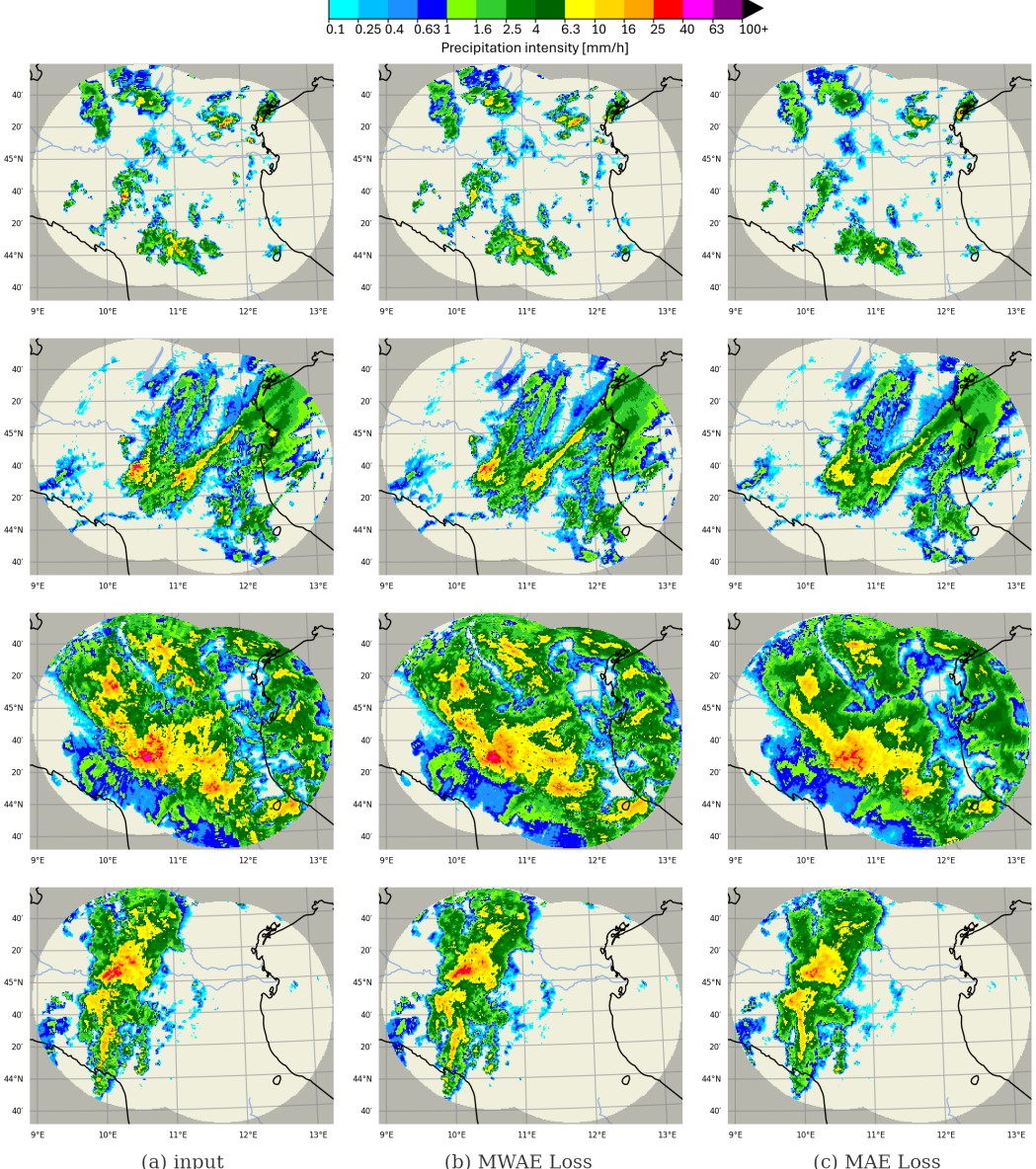

(a) input          (b) MWAE Loss          (c) MAE Loss

**Figure 8.** Qualitative comparison between precipitation snapshots reconstructed by the VQGAN autoencoder trained with MWAE loss and MAE loss, taken from the TTS. The autoencoder trained with MWAE loss shows a marked improvement in the reconstruction of precipitation, with crucial improvements in the reconstruction of higher rain rates (thunderstorms).

The last test involves an assessment of the ability of the autoencoder to reconstruct saturation-level inputs. We create a synthetic image with a saturated 64x64km patch of 205 mm/h (60 dBZ) at the center, encode it through the tokenizer, and decode the resulting token map. The reconstruction in Figure 9 visually confirms that end-of-scale values are much better

represented in the learned codebook of the MWAE autoencoder, which is able to express rain rates up to the saturation level, although not for large extents like the one provided in input. This limitation is expected due to the absence of such extensive saturated areas in the training data. Consequently, this could potentially affect the model's performance when encountering record-breaking extreme events that might exhibit such large areas of maximum intensity.

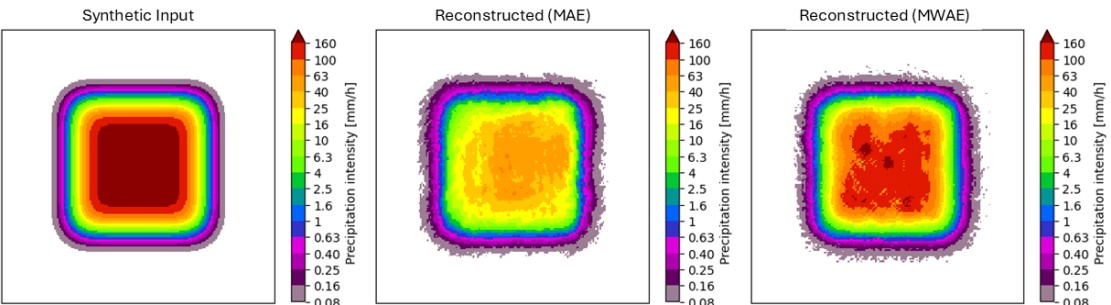

**Figure 9.** Qualitative comparison between precipitation snapshots reconstructed by the VQGAN autoencoder trained with MWAE loss and MAE loss on a synthetic saturated image. The MWAE-trained model can reach saturation-level intensities, although only over small areas.

## 4.2 GPTCast Nowcasting performance

### 4.2.1 Baseline model: LINDA

We examine and compare GPTCast forecasting performance with that of the Lagrangian INtegro-Difference equation model with Autoregression (LINDA) (Pulkkinen et al., 2021), the state-of-the-art ensemble nowcasting model included in the pySTEPS package (Pulkkinen et al., 2019). LINDA is a nowcasting technique intended to provide superior forecast skill in situations with intense localized rainfall compared to other extrapolation methods (S-PROG or STEPS). Extrapolation, S-PROG (Seed, 2003), STEPS (Bowler et al., 2006), ANVIL (Pulkkinen et al., 2020), integro-difference equation (IDE), and cell tracking techniques (Dixon and Wiener, 1993) are all combined in this model.

### 4.2.2 Verification Scores

For verification assessment, we rely on the Continuous Ranked Probability Score (CRPS) and the rank histogram, which are essential tools for verifying ensemble forecasts. By showing the frequency of observed values among the forecast ranks, the rank histogram evaluates the dispersion and reliability of ensemble forecasts and highlights biases such as under- or over-dispersion. By comparing the prediction's cumulative distribution function to the actual value, CRPS calculates a numerical score for forecast skill that indicates how accurate a probabilistic forecast is. The two scores complement each other, with the CRPS providing a measure of forecast accuracy as a whole and the rank histogram emphasizing the ensemble spread and reliability.

### 4.2.3 Performance on the Forecast Test Set

We use the FTS for our main performance comparison. Out of the 10 events in FTS, 7 are convective events occurring in spring or summer, and 3 are winter precipitation events. For each event, we produce a forecast every 30 minutes, and each forecast is a 20-member ensemble forecast with 5-minute time steps and a maximum lead time of 2 hours (i.e., 24 forecasting steps) for both LINDA and GPTCast. This results in a total of 200 forecasts (20 forecasts per event) generated per model. For GPTCast we test both of the two model configurations, GPTCast-16x16 and GPTCast-8x8.

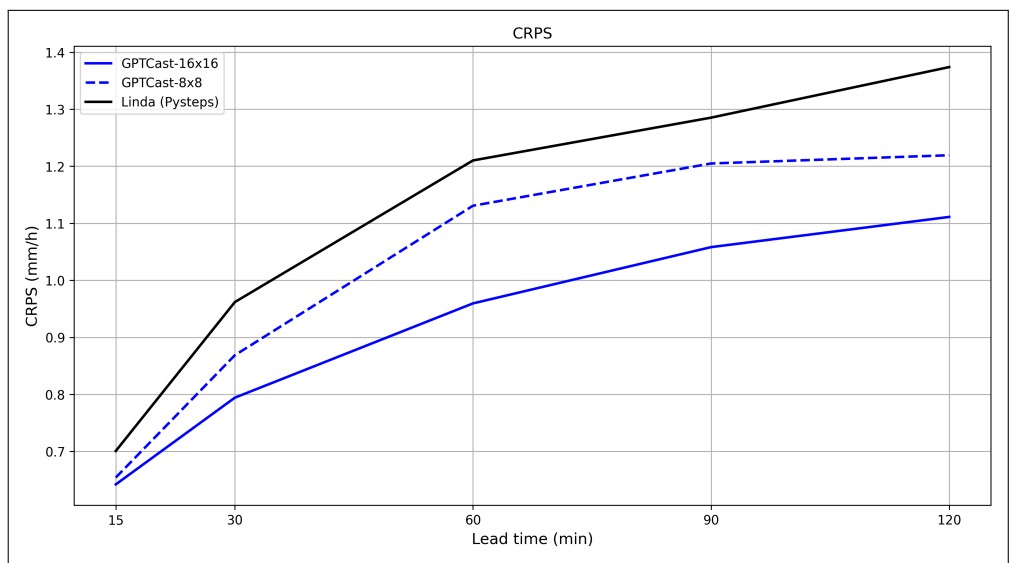

**Figure 10.** CRPS (continuous ranked probability score) comparison of GPTCast and LINDA over the FTS (lower is better) at different lead times.

The CRPS score for each of the three models—LINDA, GPTCast-16x16, and GPTCast-8x8—is displayed in Figure 10: both variants of GPTCast outperform LINDA across all lead times, with GPTCast-16x16 outperforming all other models. This result clearly shows that the model can learn a more thorough dynamic of the evolution of precipitation patterns when the context size is more spatially extended. It is important to notice that this improvement comes with a non-negligible increase in terms of computational time at inference, which in our experiments was close to an order of magnitude (GPTCast-8x8 computes a 340 timestep in 2 seconds compared to 17 seconds for the larger model on an NVIDIA RTX 4090).

Figure 11 analyzes the rank histogram at different lead times for all three models, including information on the Kullback–Leibler divergence (KL) from the uniform distribution. Both versions of GPTCast provide a better overall score than LINDA, which tends to be under-dispersed, with GPTCast-8x8 being the best model. Moreover, GPTCast-8x8 shows a rank distribution close to optimal up to the first hour, with a KL divergence from the uniform distribution of 0.006 at 60 minutes lead time (12 345 steps). GPTCast-16x16 displays an overall better rank histogram than LINDA up to the first 60 minutes with a tendency to underestimate that compounds over time: we attribute this behavior to the increased ability of the GPTCast-16x16 to capture

the training distribution, that has a higher ratio of dissipating precipitation events than the FTS (which is filtered to contain only extreme events).

Figure 12 shows an example of nowcast for a convective case in the FTS, with two ensemble members and the ensemble
mean for both LINDA and GPTCast. GPTCast generates two realistic and diverse forecasts, with an ensemble mean that features a better location accuracy than LINDA compared to the observations.

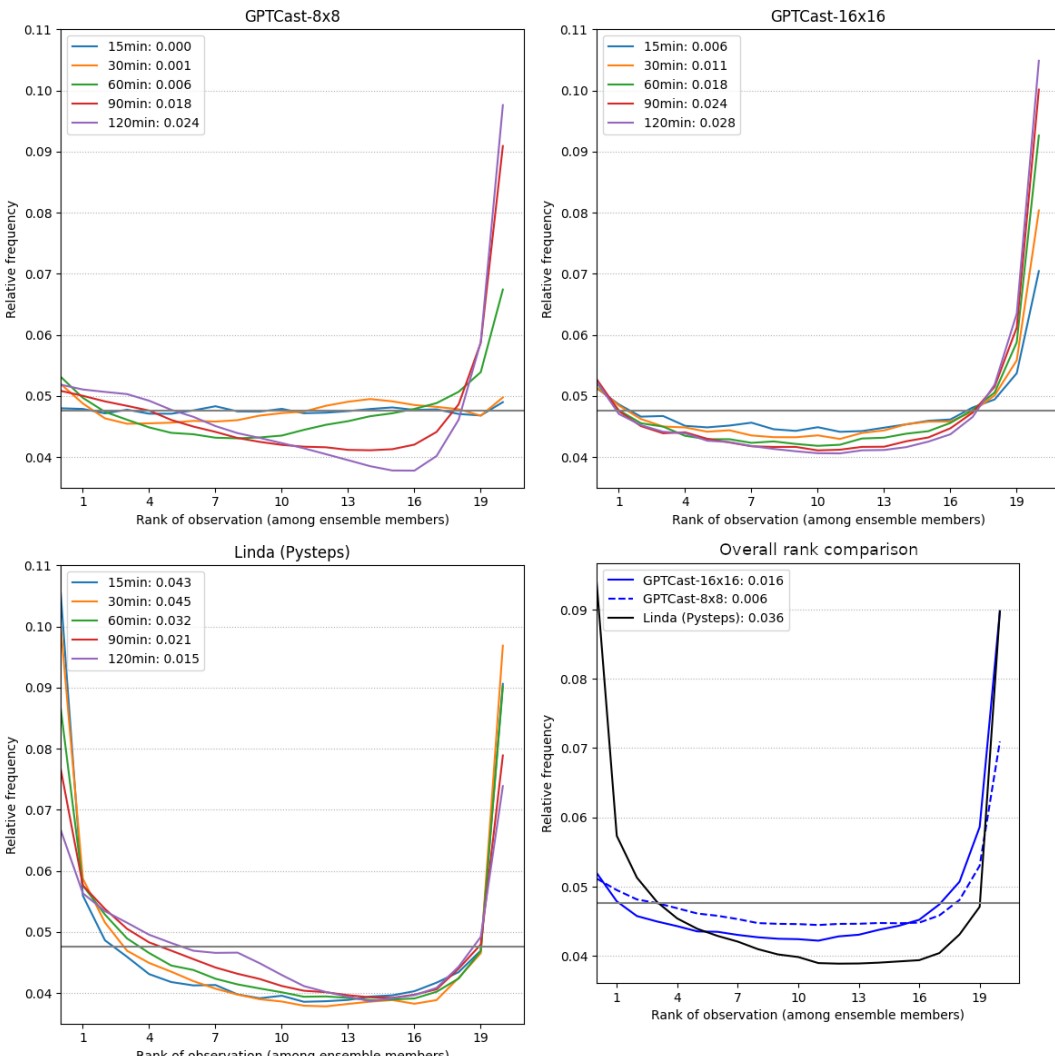

**Figure 11.** Rank histrograms comparison of GPTCast and LINDA on the FTS. The horizontal gray line represents the ideal value (the closer the better). The numbers in the legend indicate the Kullback–Leibler divergence from the uniform distribution (lower is better).

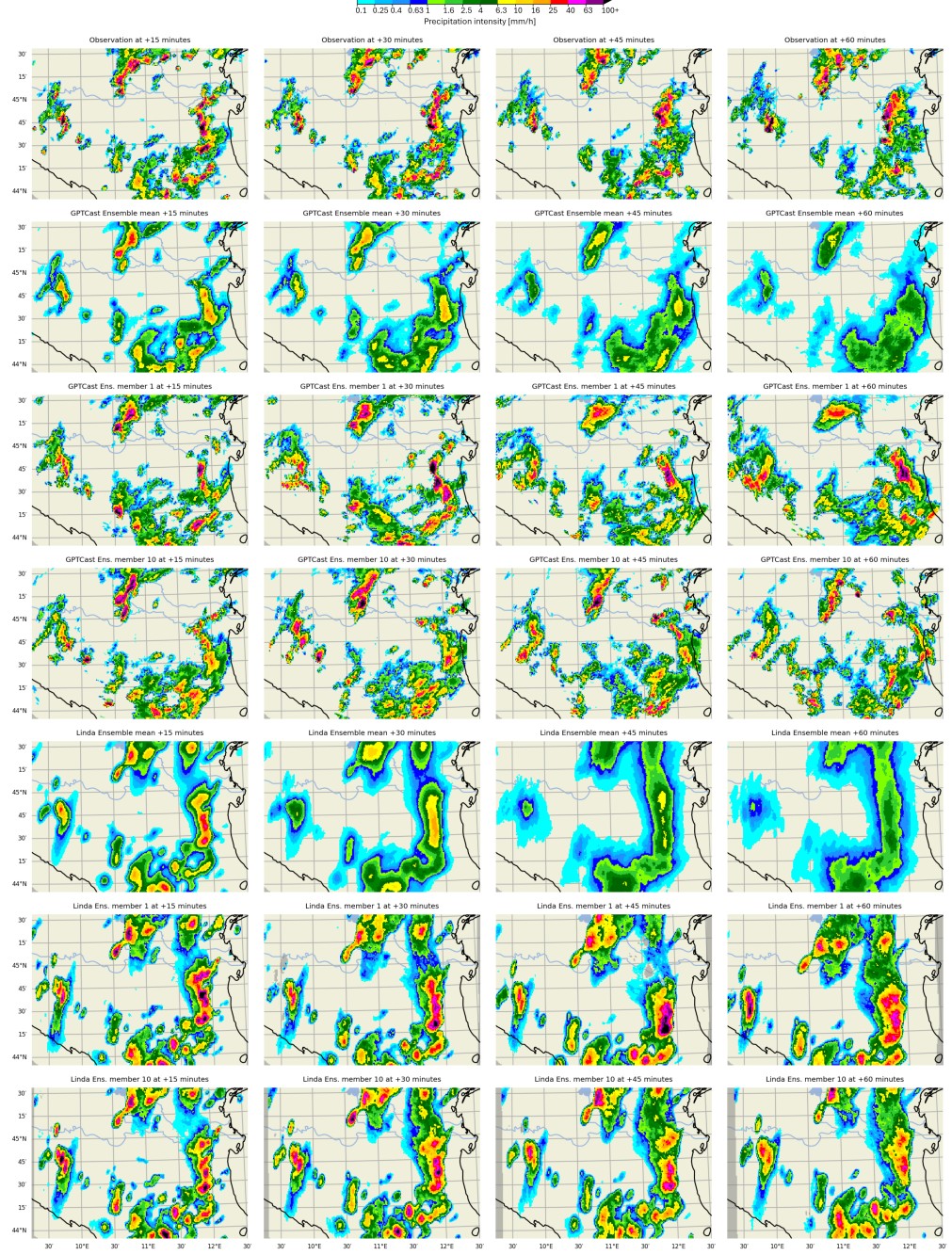

**Figure 12.** Example comparison of GPTCast-16x16 and LINDA nowcast on a convective case in the Forecaster Test Set (2020-06-08 11:00 UTC). The domain is cropped on the central area for visualization convenience.

### 4.2.4 Out-of-Distribution Evaluation on German Radar Data

To assess the generalization capability of GPTCast beyond the primary dataset used for training and testing, we perform an additional evaluation on an independent dataset from a different geographical region and source. We utilized the radar dataset over Germany presented alongside RainNet (Ayzel et al., 2020). From the first 150,000 timesteps available in this dataset, we selected the 10 cases exhibiting the highest domain-average precipitation to focus on challenging forecasting scenarios.

For each selected case, we extracted the central 256x256 pixel domain, matching the spatial dimensions used in our primary experiments. We then generated 60-minute precipitation forecasts using a 20-member ensemble for both GPTCast (specifically, GPTCast-16x16) and the LINDA.

The results indicate that GPTCast achieves a lower (better) average CRPS compared to LINDA over these 10 selected cases, suggesting better overall probabilistic forecast skill in this out-of-distribution setting. However, the rank histogram for GPTCast still exhibited a tendency towards lower ranks, consistent with the underestimation characteristic observed in the primary evaluation (Section 4.2.3).

It is important to interpret these results with caution. Firstly, the evaluation comprises only 10 cases, which limits the statistical significance of the findings. Secondly, as noted by Ritvanen et al. (2025), LINDA's performance is often optimized for and excels during high-intensity convective events. The case selection based on domain-average precipitation might not perfectly align with the scenarios where LINDA demonstrates its peak performance relative to other models. Nonetheless, this preliminary out-of-distribution evaluation provides encouraging evidence that the precipitation dynamics learned by GPTCast possess a degree of transferability to different geographical regions and data sources.

### 4.2.5 Behavior with non-precipitating input

To address the model's behavior when presented with input sequences entirely devoid of precipitation —a scenario excluded during training— we conduct an additional experiment using synthetic data. We initialized the GPTCast-16x16 model with an input sequence consisting entirely of zero-value radar reflectivity images (representing 'all clear' conditions) across the 256x256 pixel domain for the standard 7-timestep context window. We then generated an ensemble forecast of 20 members for the next time step.

The results show that most ensemble members correctly predicted continued zero (or near-zero) precipitation, consistent with a persistence forecast expected under such conditions. However, in particular, one ensemble member generates a significant spurious, albeit localized and physically plausible-looking, precipitation pattern. This highlights a potential drawback of the generative nature of the model: the possibility of "hallucinating" precipitation features when initialized with data far outside its training distribution (i.e., entirely empty sequences). While infrequent in this test (one member out of 20), this behavior warrants consideration for operational deployment and is discussed further in Section 5. Figure 14 illustrates the behavior of the members and the generated pattern from the deviating ensemble member.

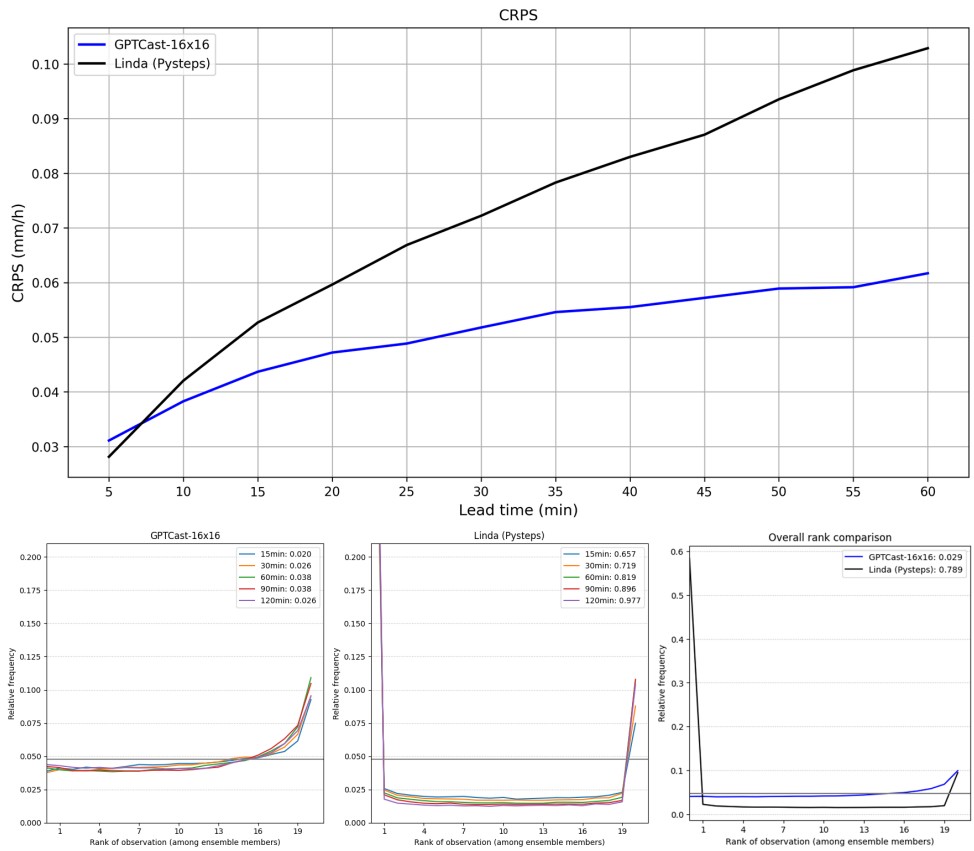

**Figure 13.** CRPS and Rank Histogram of GPTCast-16x16 and LINDA on 10 precipitation events over central Germany. GPTCast

## 5 Discussion and future work

### 5.1 Summary and Contributions

GPTCast introduces a novel approach to ensemble nowcasting of radar-based precipitation, leveraging a GPT model and a specialized spatial tokenizer to produce realistic and accurate ensemble forecasts. We show that this approach can provide reliable forecasts, outperforming the state-of-the-art extrapolation method in both accuracy and uncertainty estimation.

GPTCast's deterministic architecture enhances interpretability and reliability by generating realistic ensemble forecasts without random noise inputs. The model can be scaled to different sizes, both in context length and in terms of parameters

(which we postponed to future analyses) allowing to balance the trade-off between accuracy and computational demands and providing flexibility for different operational settings.

We believe that our method, by adopting an architecture influenced by large language models (LLMs), paves the way for future promising research in precipitation nowcasting that can incorporate all the improvements and developments from the

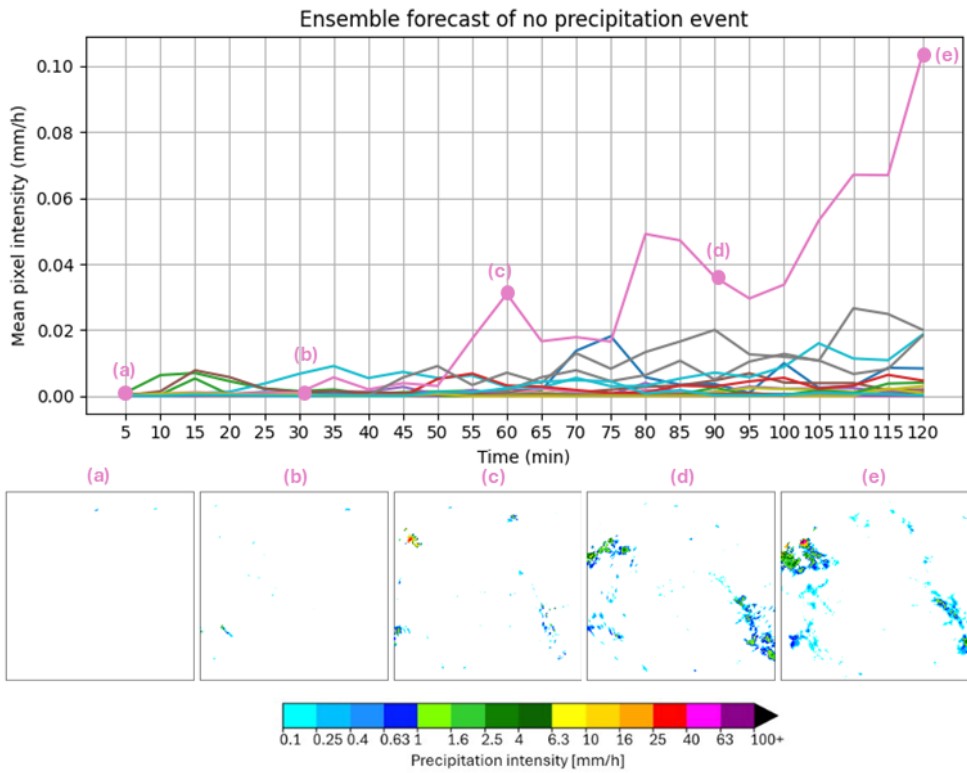

**Figure 14.** Behavior of the model initialized from a zero precipitation input sequence of 256x256 pixel domain. One out of the 20 ensemble members develops a significant precipitating pattern.

quickly developing field of LLM research. This includes more efficient architectures, improved training techniques, and better interpretability tools. Such integration can potentially enhance GPTCast's performance, scalability, and usability, ensuring that it remains a state-of-the-art nowcasting tool.

## 5.2 Implementation Challenges

Despite its strengths, the approach poses specific challenges that must be considered for the operational usage of the model.

The approach requires training two models in cascade, each with its own set of challenges. In our experiments, it was hard to find a stable configuration to train the spatial tokenizer that has to balance multiple competing losses. The MWAE reconstruction loss we introduced helped substantially in terms of both convergence and stability, although at the cost of slower training induced by the smoothing effect of the sigmoid ($\sigma$) terms in the loss. On the other hand, we found the forecaster to be very stable in training (as expected by transformers) but computationally intensive in inference, especially for the long context configuration (GPTCast-16x16), making its use in a real-time application like nowcasting challenging without significant resources.

## 5.3 Handling Non-Precipitating Conditions and Generative Artifacts

The ability of the model to effectively capture the training distribution is both its main strength and potential pitfall. A key aspect of our training strategy was the exclusion of entirely non-precipitating sequences, representing a significant portion (71.5%) of the raw data. This decision aimed to focus the model's learning capacity on the core challenge: capturing the complex dynamics of precipitation initiation, evolution, and decay, rather than diluting the learning signal with vast amounts of 'all clear' data. Operationally, if the recent radar sequence shows no precipitation, a simple persistence forecast (predicting continued 'no precipitation') is often sufficient and computationally inexpensive for the very short term, making the deployment of a complex model like GPTCast potentially wasteful in such specific situations. Our training strategy thus aligns with a targeted use case where the model is primarily invoked when precipitation is present or developing.

However, this raises the question of how the model behaves when presented with the non-precipitating inputs it might encounter operationally. While the model learns to handle the cessation of precipitation within partly precipitating sequences present in the training data, its behavior on entirely clear inputs was not explicitly trained. Our analysis in Section 4.2.5, using synthetic all-zero inputs, showed that while the model predominantly predicts continued clear conditions as expected, a small fraction of ensemble members (1 out of 20 in our test) can generate spurious precipitation patterns ('hallucinations'). This generative artifact, occurring when the input is significantly outside the training distribution, represents a potential drawback. While infrequent, this highlights the need for caution and potentially post-processing checks if the model were to be deployed in scenarios where it might frequently receive entirely non-precipitating inputs, or alternatively, implementing a simple check to bypass the deep learning model when inputs are non-precipitating. Further investigation could explore fine-tuning strategies or architectural modifications to mitigate such behavior, although the current targeted training approach already aligns well with typical operational workflows where nowcasting models are most crucial during active precipitation events. Moreover, strategies exist to exert more control over the generation process during inference and potentially reduce the occurrence of undesirable outcomes. One common technique, adapted from natural language processing, is *top-k sampling* (Fan et al., 2018; Holtzman et al., 2020). Instead of sampling from the entire probability distribution over the VQGAN codebook indices predicted by the Transformer, top-k sampling restricts the selection pool to only the $k$ tokens (codebook indices) with the highest predicted probabilities at each step. By filtering out low-probability options, this can make the generated sequences more focused and less likely to contain highly improbable or spurious transitions. However, this comes at the cost of potentially reduced forecast diversity and the risk of suppressing genuinely rare but physically valid meteorological events. Choosing an appropriate value for $k$, or exploring related techniques like nucleus sampling (top-p) (Holtzman et al., 2020), involves a trade-off between forecast creativity/diversity and robustness against potential hallucinations. Further investigation into optimal decoding strategies for precipitation nowcasting with GPTCast, possibly incorporating physical constraints or adaptive sampling methods, remains an area for future research to enhance reliability for operational use.

## 5.4 Geographical Generalizability

A further consideration regarding the generalizability of GPTCast pertains to the geographical scope of the data used for training and primary evaluation. Our main experiments were conducted using radar data covering the Emilia-Romagna region, which possesses distinct topographical features and precipitation characteristics. Consequently, the model's performance might differ when applied to regions with significantly different environments, such as coastal areas or large flat plains, which exhibit distinct precipitation regimes or atmospheric dynamics.

To provide an initial assessment of the model's robustness beyond its training domain, we performed an additional evaluation on a completely independent dataset comprising recent precipitation events over Germany, a region with different geographical characteristics (as detailed in Section 4.2.4). The promising results obtained in this out-of-distribution setting (Section 4.2.4) suggest that GPTCast learns representations of precipitation dynamics that possess some degree of geographical transferability. While these findings are encouraging, they represent only a first step. More extensive validation across a wider variety of geographical regions and climatological conditions would be necessary to fully establish the broad applicability and potential regional biases of the model, representing an important avenue for future research.

## 5.5 Inference Efficiency and Optimization Strategies

Another important practical consideration for deploying large autoregressive transformer models like GPTCast in operational settings is their computational cost during inference. While powerful, the attention mechanism and the sheer number of parameters can lead to significant latency and memory requirements. However, the field has developed numerous optimization techniques specifically targeting these challenges, which could be applied to GPTCast to enhance its real-time feasibility.

One major advancement is the development of optimized attention algorithms, such as FlashAttention (Dao et al., 2022), which reduces the memory footprint and increases the speed of the attention computation by avoiding materialization of the large attention matrix. Furthermore, model quantization techniques (Gholami et al., 2021) can significantly reduce the model size and accelerate inference by representing weights and activations using lower-precision integer formats (e.g., INT8) instead of floating-point numbers, often with minimal impact on predictive performance. Relatedly, inference can be performed using reduced precision formats like FP8 (Noune et al., 2022), which speeds up matrix multiplications on hardware accelerators supporting these formats. For autoregressive generation, efficiently managing the key-value (KV) cache is crucial (Pope et al., 2023); techniques optimizing KV cache storage and retrieval avoid redundant computations for previously processed tokens, drastically speeding up the generation of subsequent forecast steps. While the implementation and evaluation of these optimizations are beyond the scope of this initial study, their successful application in other domains suggests that they represent a viable path towards deploying models like GPTCast efficiently in time-critical operational nowcasting workflows.

## 5.6 Future Work and Outlook

Finally, as future studies, we also plan to explore the interpretability of the model to control and condition the model for different tasks. The peculiar characteristics of GPTCast open the possibility of guiding the generative process of the model

by combining the probabilistic output of the forecaster with the interpretability of the learned codebook in terms of physical quantities. A possibility that we envision is to leverage GPTCast for tasks like seamless forecasting (a.k.a. blending), generation of what-if scenarios, forecast conditioning, weather generation, and observation correction capabilities.

. Data from ARPAE Emilia Romagna . The full, preprocessed dataset used for the presented experiments is available on Zenodo (Franch et al., 2024a), including the generated ensemble forecasts to reproduce the verification scores. The pretrained models are available on Zenodo (Franch et al., 2024c). A dedicated GitHub repository (https://github.com/DSIP-FBK/GPTCast) hosts the Pytorch Lightning (Falcon and The PyTorch Lightning team, 2019) code of the models described in this paper, based on the Lightning-Hydra-Template (Yadan, 2019), licensed under the MIT License. The repository also hosts the code to reproduce the images shown in this paper. GPTCast v1.0 GitHub release is archived on Zenodo (Franch et al., 2024b) and allows to download the code to reproduce the presented experiments.

. GF conceived and conceptualized the study, designed the GPTCast architecture, implemented the code and ran the experiments. GF and ET performed the analysis and verification of the results and wrote the manuscript. VP, CC, PPA provided the data, performed the data extraction, data selection and data quality control. RW performed data format conversion. All authors revised the results and reviewed the manuscript. MC supervised the study from end to end.

. The authors declare that they have no conflict of interest.

. We acknowledge CINECA Consortium for providing the GPU resources for training and running the experiments presented in this study.

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
