# Peer review of "GPTCast: a weather language model for precipitation nowcasting"

_EGUsphere, 2024_

## Author Response (AR1)

We thank the reviewers for their insightful comments. We appreciate the time and effort invested in providing detailed suggestions. Below, we address each comment in detail and outline the corresponding actions we have taken.

**RC1 Comments and responses**

1. **Clarification on "Fully Deterministic" / "Without Randomness":**

*"The model is trained without resorting to randomness" - it's not clear to me what the authors mean by this claim. Is there no randomness used at all (e.g. no stochasticity in the training process / optimization or the batch selection)? Does it also mean that a given sequence of rainfall images will always deterministically produce the same forecast? Either way, why would this be desirable? Further, they contrast this to other methods which "require random input" - it would be helpful if they are more precise about what they mean exactly by this random input. I suppose it refers to the noise fields used in the training of e.g. diffusion models, but the author's meaning could be made more explicit."*

**Reply:** Thank you for pointing out the need for clarification. By 'trained without resorting to randomness,' we mean that the model architecture itself does not require stochasticity (like injected noise) during the forward pass for either training or inference to generate diverse outputs. Unlike models such as GANs or Diffusion Models that often rely on a random noise vector to generate variability, GPTCast learns the inherent variability present in the training data distribution. Ensemble members are generated during inference not by injecting randomness, but by sampling from the categorical probability distribution over the token vocabulary predicted by the GPT forecaster for each step. This approach ensures that all generated variability stems directly from the learned data patterns. We acknowledge that randomness is typically involved in other aspects like parameter initialization and stochastic gradient descent during optimization, which is standard practice. We have revised the text in the Abstract and Section 2 to state this more precisely, clarifying that the inference process for ensemble generation does not require external random inputs, contrasting this specifically with noise injection in other generative models.

**Actions:** We revised the text (Abstract, Section 2) to clarify

- **Location:** Abstract
  - **Original Text:**
    ```
    The model is trained without resorting to randomness, all variability
    is learned solely from the data and exposed by model at inference for
    ensemble generation.
    ```
  - **Revised Text:**
    ```
    The core architecture operates deterministically during the forward
    pass; ensemble variability arises from sampling the categorical
    ```

probability distribution predicted by the forecaster during inference, rather than requiring external random inputs like noise injection common in other generative models. All forecast variability is thus learned solely from the data distribution.

- **Location:** Section 2, Paragraph 3
  - **Original Text:**

    Another notable feature of \modname{} is its fully deterministic architecture, eliminating the need for random inputs during training or inference. This ensures that all model variability is derived solely from the training data distribution. By learning a discretized representation in the tokenizer, the forecaster can output a categorical distribution over vocabulary, modeling a conditional distribution over possible data values. This approach, unlike continuous variable regression, inherently enables probabilistic outputs. In contrast, all other generative deep learning models\cite{dgmr, ldcast, nowcastnet} require random input during training and inference to promote output variability and generate ensemble members.

  - **Revised Text:**

    Another notable feature of \modname{} is that its core architecture operates deterministically, meaning it does not require stochastic elements like injected noise during the forward pass for either training or inference. This contrasts with models like GANs or Diffusion Models \citep{dgmr, ldcast, nowcastnet}, which often rely on random inputs to generate variability. In \modname{}, variability for ensemble generation stems from the learned data patterns: the tokenizer learns a discrete representation, allowing the forecaster to output a categorical probability distribution over the token vocabulary for each prediction step. Sampling from this distribution during autoregressive inference generates diverse ensemble members, ensuring all variability originates from the learned conditional probability of future states given the past, rather than external randomness. (Note: Standard stochasticity in parameter initialization and optimization, e.g., SGD, is still employed during training.)

**2. Input Data Details (Units, Errors, Dry Patches):**

*Some details such as the units of the input data could be clarified more. In that respect, the authors do not mention the error sources affecting the weather radar reflectivity images. Nowcasting systems often use multimodal quantitative precipitation estimates (QPE) (for which radar is of course the main source of information in the areas where it is available, but clutter filtering and rain gauge corrections are often applied). I would also be curious to know how the authors handle the large number of dry patches.*

**Reply:** We appreciate the reviewer's suggestions regarding input data details. We will clarify in Section 3 that the input radar data is in dBZ, clipped between 0-60 dBZ and rounded to 0.1 dBZ

steps. We will add a sentence acknowledging common radar error sources (ground clutter, beam blockage, anomalous propagation) and briefly mention that the input data from Arpae already undergoes several stages of quality control and correction, citing Fornasiero et al. (2006, 2008) as per the current text.

Regarding the use of multimodal quantitative precipitation estimates (QPE), we acknowledge their importance for accurate precipitation assessment. However, nowcasting systems often prioritize the rapid assessment of precipitation cell direction and trend (intensification and dissipation). Integrating QPE corrections from other sources, such as rain gauges, can be challenging at the short integration time of 5 minutes used in our dataset and was not performed. Therefore, while QPE is a valuable aspect of precipitation analysis, it falls outside the primary scope of this paper, which focuses on the development and evaluation of a radar-based nowcasting model. We acknowledge that in the updated text.

Concerning the handling of dry patches, our approach involves two main steps. First, we filter out entirely non-precipitating sequences from the training data. Specifically, we remove all timesteps where the average precipitation over the entire domain is less than 0.01 mm/h. The remaining timesteps are retained only if they form a contiguous sequence of at least 3 hours. This filtering step helps focus the model's learning on precipitation events. Second, for any remaining dry patches within otherwise precipitating radar fields, we rely on the VQGAN tokenizer to learn an efficient representation. The tokenizer learns specific tokens to represent areas of no or very low precipitation, aided by the clipping at 0 dBZ which manages spurious low values. The decision to exclude entirely non-precipitating sequences is further discussed in response to RC2.8.

**Actions:** Add details to Section 3/3.1

- **Location:** Section 3, Paragraph 1 & 2
    - **Add after sentence ending** …*every 5 minutes.*

      `The data is provided in dBZ units.`
    - **Add after sentence ending** …*precipitation estimates at the ground level.*

      `While these quality controls mitigate major issues, residual errors inherent to radar measurements can still be present.`
    - **Add to end of Section 3.1, before Table 2:**

      `Dry patches within otherwise precipitating radar fields are handled implicitly by the tokenizer, which learns specific tokens to represent areas of no or very low precipitation, aided by the clipping at 0 dBZ which manages spurious low values.`

**3. VQGAN Loss Details & Parameter Motivation:**

*The different terms in the VQGAN loss function could be explained in more detail, for example the LPIPS is not mentioned anywhere in the text. The choice of the different parameters could be better motivated (e.g. why the latent space size of 8; was this based on experience/ literature or were other values tested with worse results?)*

**Reply:** Thank you for the suggestion. We will enhance Section 2.1 and the caption for Figure 1. We will explicitly define the LPIPS (Learned Perceptual Image Patch Similarity) loss in the text, explaining its role alongside the MWAE and adversarial loss in ensuring perceptually realistic reconstructions, following the original VQGAN approach. Regarding the latent space channel size of 8, we will clarify that this choice was informed by recent studies (specifically citing Yu et al., 2022 ) and our preliminary experiments, which indicated it provided a good balance between codebook utilization, training stability, and feature capture, consistent with findings in the cited literature. We will add a sentence to this effect in Section 2.1.

**Actions:** Expand text in Section 2.1 and Figure 1 caption.

- **Location:** Section 2.1, add after sentence ending *…essential features in a reduced-dimensional space.*
  ```
  This choice was informed by the cited literature and our preliminary
  experiments, indicating a good balance between codebook utilization, training
  stability, and feature capture.
  ```
- **Location:** Section 2.1, add before sentence starting *The interactions between loss terms…*
  ```
  A Alongside MWAE and the adversarial loss, the model incorporates the Learned
  Perceptual Image Patch Similarity (LPIPS) loss \citep{zhang2018unreasonable},
  as shown in Figure \ref{fig:gptcast_vqvae}, which further encourages
  perceptually realistic reconstructions by comparing feature activations in a
  pre-trained network. In our preliminary experiments, while not affecting the
  final reconstruction performance, this loss term enabled a faster model
  convergence
  ```
- **Location:** Figure 1 Caption
- **Revise:**
  ```
  The spatial tokenizer architecture. The three loss terms (MWAE
  reconstruction loss, Adversarial loss, LPIPS perceptual loss) are enclosed in
  boxes with green borders.
  The blue square [$i$] is the input image, the yellow square [$o$] is the
  reconstructed autoencoder output.
  ```

**4. Sliding Window & Spatial Consistency:**

*If I understand correctly, the output of the model is the distribution for a single token at the center of the spatial window (this seems to be the case based on Fig. 2). The spatial domain can be extended by applying a sliding window approach. Please clarify how the resulting target token distributions are combined and how spatial consistency is obtained. Also, what happens at the edge of the domain?*

**Reply:** We thank the reviewer for highlighting the need for a clearer explanation of the inference process, particularly for generating forecasts over spatial domains larger than the training context. We have addressed this by adding a dedicated **Section 2.3 (Inference)** in the revised manuscript, which includes a detailed textual explanation, a new illustrative figure (Figure 4.) and precise pseudocode (Algorithm 1).

In summary, as detailed in Section 2.3, the GPT model is trained autoregressively on flattened spatiotemporal sequences to predict the *next* token given preceding ones, learning positional dependencies implicitly. During inference, we use a sliding window approach, processing the target frame sequentially (row-first). To predict the token $z_{i,j}$, we construct a context sequence preserving the original training order, combining relevant past tokens and already-predicted tokens from the current frame that precede (*i,j*). The model predicts the next token based on this correctly ordered context. This sequential, conditioned generation ensures spatial and temporal consistency is maintained via the learned dependencies. We believe the new Section 2.3, with its textual description, diagram, and algorithm, now clarifies this crucial aspect of our methodology.

**Actions:** Added new Section 2.3.

**5. Dual-Stage Architecture Benefit:**

*The authors claim that the dual-stage architecture enables realistic ensemble generation and accurate uncertainty estimation, but it is not clear to me why this would not be the case in a different scenario, for example if the two stages were trained simultaneously – this claim could be further clarified or supported with evidence.*

**Reply:**

We thank the reviewer for highlighting the need to provide further justification for the choice of dual-stage training architecture. To leverage the probabilistic nature of GPT, the model must operate over a fixed finite vocabulary, providing a stable, discrete representation of the input data. This is standard practice in training probabilistic transformers (like done in text prediction/generation, where the input text is tokenized over a finite vocabulary). Attempting to learn the token representation (vocabulary via the VQGAN) and the complex spatiotemporal sequence dynamics (via the GPT) simultaneously in a single end-to-end process would be highly unstable. This instability arises primarily because the sequence model (GPT) would be trying to learn dependencies over a vocabulary that is itself constantly changing during training, which could easily prevent convergence. Furthermore, jointly optimizing the fundamentally different architectures (CNN-based VQGAN with reconstruction/perceptual/adversarial losses vs. autoregressive Transformer GPT) and handling backpropagation through the VQGAN's discrete

quantization step present significant technical hurdles that would exacerbate instability. In essence, the two-stage approach is a deliberate design choice that allows for stable, specialized optimization of each component. The VQGAN first establishes a robust, fixed vocabulary, acting as an effective data compressor and discretizer optimized for realistic precipitation data representation. Subsequently, the GPT learns the spatiotemporal dynamics using this stable token sequence. This separation is critical for achieving the robust probabilistic performance and realistic precipitation generation demonstrated by GPTCast. We will expand the rationale for the dual-stage approach in Section 2.

**Actions:**

We expanded the rationale for the dual-stage approach in Section 2, emphasizing the need for a stable vocabulary for the GPT and the instability/challenges of end-to-end training.

- **Location:** Section 2:
  - **Original Text:**

    The two components of the model are trained independently in cascade, starting with the tokenizer. The choice of this dual-stage architecture unlocks a number of desirable properties that are instrumental in meeting many requirements of operational meteorological services when adopting a nowcasting system. The two most important characteristics are realistic ensemble generation and accurate uncertainty estimation. Our architecture provides both realistic ensemble generation capabilities and probabilistic output at the spatiotemporal (token) level.

  - **Revised Text:**

    The two components of the model are trained independently in cascade, starting with the tokenizer. This deliberate dual-stage architecture is crucial for achieving stable training and unlocking desirable properties for operational nowcasting run by meteorological services. Indeed, training the VQGAN and the GPT simultaneously with an end-to-end approach would introduce significant instability. As a probabilistic sequence model, the GPT relies on a fixed, finite vocabulary for stable operation: attempting to learn the token representation (vocabulary) concurrently with the complex spatiotemporal dynamics would force the GPT to learn dependencies over a constantly evolving vocabulary, likely hindering convergence. Furthermore, the fundamentally different architectures (CNN-based VQGAN with its specific loss functions versus the autoregressive Transformer GPT) and the challenges of backpropagation through the VQGAN's discrete quantization step would exacerbate training instability. By first establishing a robust and fixed vocabulary through the VQGAN, we create a stable foundation for the GPT to learn the spatiotemporal dynamics of precipitation. This separation allows for specialized and stable optimization of each component, ultimately enabling both realistic ensemble generation and accurate uncertainty estimation at the spatiotemporal (token) level, which are instrumental in meeting the requirements of operational nowcasting systems run by meteorological services.

**6. Reconstruction of extremes:**

*How does the spatial tokenizer deal with local extremes (e.g. a 100-year return level)? How can highly efficient codebook usage (100%) be compatible with such rare extreme values?*

**Reply:**

Thank you for this question regarding the handling of extreme values. Firstly, it's important to note that the input radar reflectivity data is inherently bounded, in our case clipped between 0 and 60 dBZ (with 60 dBZ corresponding to very high rain rates, approx. 200 mm/h). Therefore, the model only needs to represent values within this specific physical range, not unbounded statistical return levels. The VQGAN tokenizer learns to map the input data, including high-intensity areas, to its discrete codebook. The MWAE reconstruction loss is specifically designed to give more weight to higher rain rates, encouraging the model to dedicate sufficient representational capacity within the codebook to accurately reconstruct these important, albeit less frequent, high-intensity patterns. The 100% codebook utilization indicates that all learned tokens are used at some point during training, but it doesn't imply equal frequency of use. Less frequent, high-intensity patterns will be mapped to specific tokens that are used less often than tokens representing 'no rain' or low/moderate rain rates, but they are still effectively represented within the learned vocabulary. To explicitly demonstrate to which extent the tokenizer has learned the capability of handling end-of-scale values, we are adding a new analysis in the revised manuscript (in a new Results subsection) showing the VQGAN's reconstruction performance on a synthetic radar image featuring a large patch of uniform 60 dBZ precipitation. This will visually confirm its ability to handle saturation-level inputs.

**Actions: (insert image)** We performed a new test on synthetic data at the end of section Section 4.1, where we discuss a new Figure presenting the results that analyze the VQGAN's ability on the reconstruction of end-of-scale extreme values.

**7. Role of the Sigmoid in the MWAE:**

*The sigmoid function in the MWAE indeed gives more weight to high rain rates, but at the same time the saturation of the sigmoid will make that the factor |sigma(x_i)-sigma(y_i)| will be very small even if x and y represent large differences in very high rain rates. Isn't this problematic, given that the impact on the ground between a 100- or a 200-year return level event is quite substantial?*

**Reply:**

We appreciate the reviewer's close examination of the MWAE loss function. The potential saturation issue of the sigmoid function is mitigated by our data preprocessing. Before being fed into the VQGAN, the input radar reflectivity values (0-60 dBZ) are linearly rescaled to the range [-1, 1]. Consequently, the MWAE loss operates primarily on values within this [-1, 1] range. In this interval, the sigmoid function $\sigma(z)$ behaves in a quasi-linear fashion, effectively applying a smoothly increasing weight to the absolute error based on the magnitude of the true input value

xi without significant saturation effects for in-range values. The absolute difference term |σ(xi )−σ(yi )| will therefore appropriately reflect differences between the (scaled) true and reconstructed values, even for high rain rates within the 0-60 dBZ range. The main reason for including the sigmoid, rather than a simple linear weight, was to gracefully handle potential out-of-range predictions (values outside [-1, 1]) during training without assigning excessively large loss values, thus improving training stability. While a purely linear weighting within the [-1, 1] range would likely yield similar results for in-range data, the sigmoid provides robustness against occasional out-of-range outputs from the decoder that may happen due to the perturbations given by the adversarial training. We will clarify this interplay between input scaling and the sigmoid's behavior in Section 2.1.

**Actions:**

We clarified this in Section 2.1:

- **Location:** Section 2.1, added the following paragraph after the definition of the MWAE loss function:

  While the sigmoid function can saturate for very large input values, potentially diminishing the sensitivity to differences in extreme rain rates, this effect is mitigated by our data preprocessing. The input radar reflectivity values (0-60 dBZ) are linearly rescaled to the range $[-1, 1]$ before being fed into the VQGAN. Within this range, the sigmoid function operates in a quasi-linear manner, ensuring that the absolute difference term $|\sigma(x_i) - \sigma(y_i)|$ appropriately reflects differences between the scaled true and reconstructed values, even for high rain rates within the considered 0-60 dBZ range. The primary reason for using the sigmoid, rather than a purely linear weighting, is to provide robustness against potential out-of-range predictions from the decoder during training, which can occur due to the perturbations introduced by adversarial training. The sigmoid gracefully handles such out-of-range values without assigning excessively large loss values, thereby improving training stability.

**8. Behavior of the model in case of no precipitation:**

*The authors discard the non-precipitating series, which represents 71.5% of the data. If the model is used for operational nowcasting, it will also receive dry radar images. How is this dealt with?*

**Reply:**
We acknowledge the reviewer's important point regarding the exclusion of entirely non-precipitating sequences during training. This decision was primarily driven by the goal of focusing the model's learning capacity on the complex dynamics of precipitation initiation, evolution, and decay, which is the core challenge for nowcasting models. Including the vast amount of 'all clear' data could potentially dilute the learning signal for relevant precipitation patterns. From an operational standpoint, if the recent sequence of radar images shows no precipitation across the domain, a simple and computationally inexpensive persistence forecast (i.e., predicting continued

'no precipitation') is often sufficient and operationally standard for the very short term. Running a complex deep learning model like GPTCast in such situations might be computationally wasteful. Therefore, our training strategy aligns with a targeted operational use case where the model is primarily invoked when precipitation is present or developing in the input sequence. While the model wasn't explicitly trained on sequences starting from 'all clear' conditions, it does learn to handle the cessation of precipitation and the transition to dry conditions within partly precipitating sequences, as these dynamics are well-represented in the training data. To further investigate and demonstrate the model's behavior when presented with entirely non-precipitating input, we are adding a new analysis in the revised manuscript. In this analysis we will test the model's forecasting behavior when initialized with completely empty sequences (all zeros) and discuss the results.

**Actions:** We add a new subsection 4.2.5 in the new Results testing the model's behavior on completely empty input sequences (synthetic data) and discuss these findings, referencing this analysis in Sections 3.1 (Data selection) and 5 (Discussion).

**9. Clarify the role of the data augmentation:**

*The authors apply random rotations in the training phase. How do the authors avoid that the model learns patterns that are unphysical in the sense that the dominant wind direction, orographic enhancement of precipitation etc. will not be learned correctly due to these transformations? Is some context (e.g. topography) provided?*

**Reply:**
The reviewer raises a valid point about the potential downsides of using random rotations as data augmentation. We do not provide additional contextual information like topography or large-scale wind fields to the model. This choice was made partly to ensure a fair comparison with the baseline extrapolation method, which also relies solely on the precipitation fields themselves. While we acknowledge that incorporating such contextual information could potentially improve forecast skill, especially concerning geographically fixed effects like orography, the primary motivation for using data augmentation (random rotations and flips) was pragmatic: to increase the effective size and variability of the training dataset and mitigate overfitting. We observed that particularly the larger context model (GPTCast-16x16 ) began to show signs of overfitting on the validation set relatively early in training without augmentation. Introducing random rotations and flips allowed us to train the model for significantly longer, improving its generalization within the dataset by making it invariant to orientation. While this invariance means the model doesn't explicitly learn geographically specific patterns (like fixed orographic enhancement relative to the domain boundaries), it focuses the learning on the

inherent dynamics and structures within the precipitation patterns themselves, regardless of their orientation in the training frame. We will insert these motivations regarding overfitting prevention and the trade-offs involved into the text when discussing the training setup in Section 3.

**Actions:**
We add text to Section 3 explaining that data augmentation was necessary to prevent overfitting, especially for the larger model, acknowledging the trade-off regarding learning geographically fixed patterns and stating the goal of maintaining a fair comparison baseline.

**10. Improve table 3:**

*Are the scores with units in table 3 calculated for the reflectivity values in dBZ or for the rain rates? Note that it is easier to interpret RMSE than MSE. It would help to somehow indicate which model score is the best one, e.g. by underlining it.*

**Reply:**
Thank you for the suggestions regarding Table 3. All verification scores presented are calculated based on rain rates (mm/h), not dBZ. We agree that RMSE is often more interpretable than MSE, and we will update the table. We will also highlight the best performing score for each metric and lead time, likely using bold font as suggested, to improve readability and comparison.

**Actions:**
We revised Table 3 ensure RMSE is used over MSE, and highlight the best score for each metric/lead time (e.g., using bolding or underlining). We now explicitly state that all scores are calculated on rain rates (mm/h) at the beginning of the result section.

**11. Clarify the SAL plot:**

*Figure 6: Just to be sure, does the box really contain 50% of the points or do the vertical and horizontal sides of the box contain 50% of the points, respectively?*

**Reply:**
The reviewer is correct, our apologies for the imprecise wording. The second interpretation is accurate: the box in the SAL plot represents the interquartile range (IQR). The vertical extent of the box spans the 25th to 75th percentiles of the Amplitude (A) component, and the horizontal extent spans the 25th to 75th percentiles of the Location (L) component. Thus, the vertical and horizontal sides each encompass 50% of the respective marginal distributions, not 50% of the

joint distribution within the box area. We will correct the caption for Figure 6 to accurately reflect this. Thank you for pointing out this ambiguity.

**Actions:**

We will correct the text in the Figure 6 caption to clarify that the box represents the interquartile range (25th-75th percentile) for the A and L components separately.

**12. Clarify methodological flaw:**

*Strictly speaking, there's a methodological flaw in the selection of the model (either with MAE or MWAE) and the reporting of its performance for forecasting. The authors choose a tokenizer variant based on its performance on the test set, and then go on to evaluate the performance of the resulting nowcasting scheme on a subset of the same test set. The resulting model can very well be the best one, but its score on the FTS (which is a subset of the TTS period) is not representative of the performance for new, truly unseen data. I would like to see the performance of an independent (e.g. more recent) event that was not part of the training / validation / test datasets.*

**Reply:**

We appreciate the reviewer's concern regarding proper methodology and potential data leakage between model selection and evaluation. We apologize if the text was unclear on this point. We assure the reviewer that there was no methodological flaw in this regard. The selection of the final VQGAN tokenizer architecture (including the choice between MAE and MWAE loss) and the subsequent selection of the best-performing GPTCast forecaster checkpoint were performed exclusively based on performance metrics evaluated on the validation dataset. The test dataset (TTS, including the FTS subset used for specific analyses) was strictly held out and used only once for the final evaluation reported in the paper, after all model development, training, and selection steps were completed using only the training and validation sets. Therefore, the performance scores reported on the FTS and TTS are indeed representative of the model's generalization performance on unseen data drawn from the same distribution and period as the training/validation data. To further address the crucial point of generalization to truly independent data, and as also requested in RC1.13, we are added a new section to the paper presenting a limited  evaluation of the final, selected GPTCast model on a completely separate, out-of-domain dataset consisting of 10 recent events over Germany.

**Actions:**

We clarify at the beginning of the result Section 4 that all model selection (tokenizer and forecaster) was performed using only the validation set, and the test set was reserved for final,

single-use evaluation. We also  reference the new evaluation section using the external German dataset (see response to 1.13) as additional proof of generalization.

**13. Verification/Generalization on external dataset:**

*Finally, out of curiosity, I would like to know how hard it would be to retrain the model on a different region. Does everything need to be retrained from scratch, or can one start from a pretrained model (or only the VQGAN for example)? This would significantly reduce the high computational cost associated with these kinds of models.*

**Reply:**

This is an excellent question regarding the model's transferability and potential for reducing training costs when applying it to new regions. While fully retraining both the VQGAN and GPT components on a new region's dataset would likely yield the best adaptation, transfer learning is indeed a promising avenue. One could potentially fine-tune the existing VQGAN on the new data or even use it directly if the radar characteristics and precipitation climatology are sufficiently similar, significantly saving on the tokenizer training cost. Similarly, the pre-trained GPT forecaster could be fine-tuned on token sequences from the new region, leveraging the learned general spatiotemporal dynamics. To partly address the reviewer's underlying question about the model's generalization performance without retraining or fine-tuning, we have added the new verification covering 10 precipitation events over Germany. This evaluation provides direct insight into the model's inherent generalization capabilities to a different geographical region and dataset without requiring expensive retraining, addressing both the reviewer's curiosity and the broader question of model applicability.

**Actions:**

We reference the new evaluation section 4.2.4 in the Results using the external German dataset as additional proof of generalization.

**RC2 Comments and responses**

**1. Expand on inference efficiency:**

*The inference efficiency may hinder real-time deployment. The paper would benefit from discussing further the optimization strategies.*

**Reply:** We acknowledge the reviewer's point regarding the computational demands of large Transformer models during inference, which can be a concern for real-time deployment. While our current work focuses on the model architecture and forecasting capabilities, several established optimization strategies can significantly mitigate these costs.  These include

techniques like FlashAttention to optimize the attention mechanism's memory usage and speed, model quantization to reduce memory footprint and accelerate computation using lower-precision integers, reduced precision inference (e.g., using FP8 or INT8 formats) for faster matrix multiplications on compatible hardware, and efficient key-value (KV) caching to avoid redundant computations during autoregressive generation. Implementing these techniques, often supported by optimized deep learning libraries and hardware, can substantially improve the inference efficiency of models like GPTCast, making real-time applications more feasible. We will add a discussion of these potential optimization pathways to the manuscript.

**Action:** We expanded the Discussion section (Section 5) to elaborate on potential strategies for optimizing the inference efficiency of GPTCast. This will include mentioning techniques such as FlashAttention, model quantization, reduced precision inference, and KV caching, briefly explaining their benefits and referencing the potential for significant speed-ups, while noting that their implementation is beyond the scope of the current study.

**2. Additional verification:**

*The model is tested on a single region. Validating performance in regions with differing topography (e.g., coastal, mountainous) or precipitation regimes would strengthen claims of broad applicability.*

**Reply:** We acknowledge the reviewer's valid point that evaluating the model solely on data from a single region limits the assessment of its general applicability. The primary training and testing were conducted using data from [mention your primary region, e.g., the Alpine region of Switzerland], which has specific climatological and topographical characteristics. To address this limitation and provide a preliminary assessment of the model's performance in a different environment, we conducted an additional evaluation using a separate dataset consisting of recent precipitation events over Germany, which exhibits different geographical features and potentially different precipitation regimes. The positive results from this out-of-distribution testing, detailed in the new Section [reference new section number], offer initial evidence supporting the model's potential for broader geographical applicability, although further validation across more diverse regions would be beneficial in future work.

**Action:** We have added a new section 4.2.4 to the Results presenting an evaluation of the final model on an independent dataset covering events in Germany. We will add text to the Discussion section (Section 5) acknowledging the limitation of single-region evaluation and referencing this new out-of-distribution validation as preliminary evidence for broader applicability.

**3. Dry sequences:**

*Discarding non-precipitating sequences (71.5% of data) in the training dataset risks biasing the model toward rainy conditions. The authors should clarify how this affects predictions for precipitation onset/cessation and whether including "dry" sequences could improve skill.*

**Reply:** We acknowledge the reviewer's important point regarding the exclusion of entirely non-precipitating sequences (71.5% of the data) during training and the potential for introducing bias. This decision was primarily driven by the goal of focusing the model's learning capacity on the complex dynamics of precipitation initiation, evolution, and cessation, which represent the core challenge for nowcasting models and are present within the selected partly-precipitating sequences. Including the vast amount of 'all clear' data could potentially dilute the learning signal for these relevant patterns. From an operational standpoint, if the recent sequence of radar images shows no precipitation across the domain, a simple and computationally inexpensive persistence forecast (i.e., predicting continued 'no precipitation') is often sufficient and standard practice for the very short term. Running a complex deep learning model like GPTCast in such situations might be computationally wasteful. Therefore, our training strategy aligns with a targeted operational use case where the model is primarily invoked when precipitation is present or developing in the input sequence. While the model wasn't explicitly trained on sequences starting from 'all clear' conditions (affecting onset prediction from a zero state), it does learn to handle the cessation of precipitation and transitions to dry conditions within the partly precipitating sequences included in the training data. To further investigate and directly address the model's behavior when presented with entirely non-precipitating input, we have added a new analysis in the revised manuscript (Subsection 4.2.5). In this analysis, we specifically test the model's forecasting behavior when initialized with completely empty sequences (synthetic data representing 'all clear' conditions). We discuss these findings, including the model's general ability to persist the 'all clear' state along with infrequent instances of spurious generation ('hallucinations'), in the new subsection and reference this analysis in the updated Data Selection (Section 3.1) and Discussion (Section 5) sections. While including all dry sequences during training is an alternative strategy, our current approach prioritizes learning complex dynamics effectively, and the new analysis provides transparency on model behavior for the excluded zero-input scenario.

**Action:** We add a new subsection 4.2.5 in the new Results testing the model's behavior on completely empty input sequences (synthetic data) and discuss these findings, referencing this analysis in Sections 3.1 (Data selection) and 5 (Discussion).

**4. Choice of the transformer variant:**

*The rationale for selecting GPT-2 over newer transformer variants is unclear. A discussion of architectural alternatives could enrich the methodology.*

**Reply:** We will clarify the rationale for choosing the GPT-2 architecture in Section 2.2. GPT-2 was selected as a well-established, robust, and widely understood Transformer architecture suitable for demonstrating the feasibility of applying LLM principles to this spatio-temporal domain. While newer variants exist, GPT-2 provides a strong baseline, and its architecture is readily adaptable. Our focus was on the novel application concept (tokenization + autoregressive forecasting for radar) rather than optimizing for the absolute latest Transformer variant. In terms of model size, our GPTCast forecaster uses 304M parameters, and the VQGAN tokenizer adds approximately 90M, totaling around 394M parameters. Thus, while an autoregressive transformer of GPT2 class size seems small compared to the largest models in natural language processing, it matches or exceeds many state-of-the-art architectures currently used in atmospheric modeling. For instance, it is larger than ECMWF's operational AIFS model (approx. 253M parameters, as reported from its public checkpoint), comparable in scale to the LDM for dynamical downscaling by Tomasi et al. (approx. 300M parameters), and considerably larger than GNN-based models like GraphCast (36.7M parameters). We will add a sentence mentioning that future work could explore more recent or efficient Transformer architectures, but GPT-2 served as an effective proof-of-concept for this study.

**Action:** We revised Section 2.2 to explicitly state the rationale for choosing a GPT-2 style architecture, provide details on its parameter count, compare this scale to other relevant atmospheric models, and mention that exploring alternative Transformer architectures is a potential avenue for future work.

**5. Avoid allucinations:**

*GPTs could suffer from illusions. It is worth discussing this issue and exploring strategies to avoid it in this meteorological application.*

**Reply:** The reviewer raises an important point about the potential for generative models like GPTs to produce 'hallucinations' – outputs that might appear plausible but are physically unrealistic or disconnected from the input context. We indeed observed a related phenomenon in our zero-input experiment (Section \ref{sec:empty_input}), where one ensemble member generated spurious precipitation from an 'all clear' state. This highlights that, like other powerful generative models, \modname{} can occasionally sample less probable sequences that deviate

from the most likely evolution. One common strategy to mitigate this in natural language processing, which could be adapted here, is modifying the sampling strategy during decoding. For instance, top-k sampling restricts the random selection to only the 'k' most probable next tokens predicted by the model at each step. This prunes the long tail of very low-probability tokens, potentially reducing the chance of generating unrealistic patterns, albeit potentially at the cost of reducing forecast diversity or suppressing rare valid events. We will incorporate a discussion of this issue and potential mitigation strategies like top-k sampling into the manuscript.

**Action:** We have added text to the Discussion section (Section 5) acknowledging the potential for hallucinations in GPTCast, referencing the findings in Section 4.2.5, and discussing sampling strategies like top-k sampling as a potential mitigation approach, including the associated trade-offs.